# Sparse maximal update parameterization: A holistic approach to sparse training dynamics

**Nolan Dey**   **Shane Bergsma**   **Joel Hestness**
Cerebras Systems
{nolan,joel}@cerebras.net

## Abstract

Several challenges make it difficult for sparse neural networks to compete with dense models. First, setting a large fraction of weights to zero impairs forward and gradient signal propagation. Second, sparse studies often need to test multiple sparsity levels, while also introducing new hyperparameters (HPs), leading to prohibitive tuning costs. Indeed, the standard practice is to re-use the learning HPs originally crafted for dense models. Unfortunately, we show sparse and dense networks do not share the same optimal HPs. Without stable dynamics and effective training recipes, it is costly to test sparsity at scale, which is key to surpassing dense networks and making the business case for sparsity acceleration in hardware.

A holistic approach is needed to tackle these challenges and we propose sparse maximal update parameterization (SμPar) as one such approach. For random unstructured static sparsity, SμPar ensures activations, gradients, and weight updates all scale independently of sparsity level. Further, by reparameterizing the HPs, SμPar enables the same HP values to be optimal as we vary both sparsity level and model width. HPs can be tuned on small dense networks and transferred to large sparse models, greatly reducing tuning costs. On large-scale language modeling, SμPar shows increasing improvements over standard parameterization as sparsity increases, leading up to 11.9% relative loss improvement at 99.2% sparsity. A minimal implementation of SμPar is available at https://github.com/EleutherAI/nanoGPT-mup/tree/supar.

## 1 Intro

*Sparsity* has emerged as a key technique to mitigate the increasing computational costs of training and inference in deep neural networks. This work focuses on *weight sparsity*, whereby a significant fraction of model weights are kept at zero. It has long been known that dense neural networks can be heavily pruned *after* training [30]. With the goal of reducing costs *during* training, recent work has explored static weight sparsity from initialization. In this work we focus on random unstructured static sparsity, which has re-emerged as a surprisingly effective strategy [33, 58]. This type of sparsity can be accelerated by CPUs, Cerebras, Graphcore, and SambaNova. Furthermore, GPUs and TPUs support 2:4 block structured sparsity which is quite similar to 50% unstructured sparsity.

Unfortunately, several challenges have hindered progress in weight-sparse neural networks. First, sparsity impairs signal propagation during training [31, 11, 1]. Second, with today's techniques, sparse training is costly. Sparse techniques typically introduce extra hyperparameters (HPs), e.g., number of pruning iterations at initialization [60, 7, 56], and it is common to train models across different sparsity levels. Since tuning should be performed at each level and the search space grows exponentially with the number of HPs, the tuning costs essentially "defeat the purpose" of sparsity, i.e., to *reduce* computation [60]. Finally, today there is only a nascent ecosystem of hardware acceleration for unstructured sparsity, so most researchers get little sparsity benefit when tuning.

38th Conference on Neural Information Processing Systems (NeurIPS 2024).

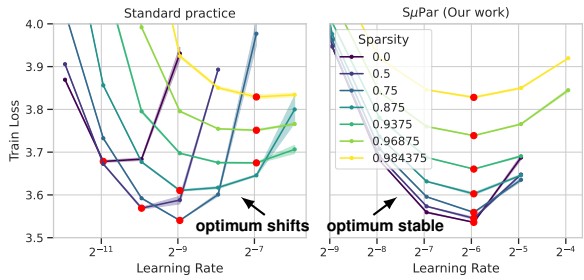

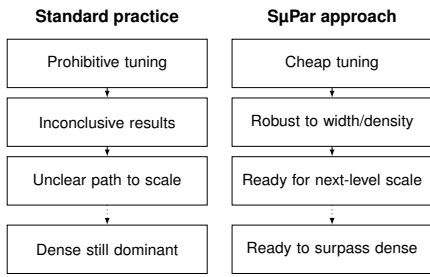

Figure 1: SµPar (Our work) allows stable optimum HPs for any sparsity level, unlike standard practice.

Figure 2: SµPar enables sparse training at scale, helping to surpass dense and motivate sparsity in hardware.

These costs have led to the standard practice of *simply re-using HPs that were previously optimized for the baseline dense models* (Section 2). One might hope that sparse models thrive with the same learning rates and other HPs as their dense counterparts. Unfortunately, they do not: optimal HPs *systematically* vary with sparsity level (Figure 1, left). With impaired training dynamics, prohibitive tuning cost, and lacking the established training recipes enjoyed by dense models, it is often inefficient to train sparse networks at scale (Figure 2).

To remedy this situation, we propose sparse maximal update parameterization (SµPar, pronounced "soo-pahr"), a novel, holistic approach to stabilize sparse training dynamics. SµPar fulfills the Feature Learning Desiderata (Section 3) by parameterizing weight initialization and learning rates with respect to change in width *and* sparsity level. As a generalization of maximal update parameterization (µP) [64, 63], SµPar enjoys well-controlled activation, gradient, and weight update scales in expectation, avoiding exploding or vanishing signal when changing both sparsity and model width.

By reparameterizing HPs in this way, SµPar enables the same HP values to be optimal as sparsity varies (Figure 1, right). We therefore enjoy µTransfer: we can tune small proxy models and transfer optimal HPs directly to models at scale. In fact, we discovered our µP HPs, tuned for dense models in prior work (and equivalent to SµPar with sparsity=0%), correspond to the optimal learning rate and initial weight variance for *all* sparse models tuned in this paper! As sparsity increases, our formulation shows the standard parameterization (SP) and µP suffer from vanishing signal, further clarifying prior observations of gradient flow issues in sparse networks. The improvements enabled by SµPar set the Pareto-frontier best loss across sparsity levels. Figure 3 previews this improvement for large language models trained from compute-optimal configurations [23]. Here, SµPar benefits grow with increasing sparsity, to 11.9% better loss than SP and 1.9% better loss than µP at 99.2% random unstructured sparsity. See Section 4.3 for details on this experiment.

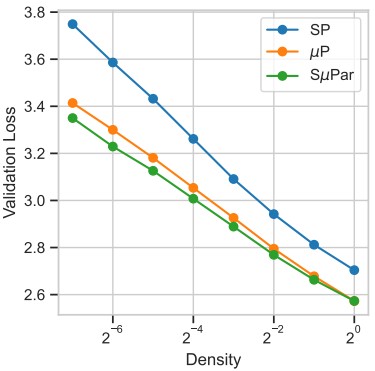

Figure 3: For LLMs, SµPar forms the Pareto frontier loss across sparsity levels, with no HP tuning required.

## 2 Related work

**Sparse training landscape** Sparse training can be divided into static sparsity, where the connectivity is fixed (our focus) and dynamic sparsity, where the sparsity mask can evolve [22]. We use *unstructured* sparsity, though our approach generalizes to structured approaches where a particular sparsity pattern increases efficiency on specific hardware [67, 26, 38, 14, 29, 1]. Unstructured connectivity may be based on both random pruning [40, 18, 57, 33, 58] and various pruning-at-initialization criteria [32, 60, 61, 56, 7]. Liu et al. [33] found that as models scale, the relative performance of randomly pruned networks grow. Furthermore, Frantar et al. [15] found the optimal level of sparsity increases with the amount of training data. Together, these findings suggest that as neural networks

continue to get wider and deeper, and trained on more and more data, very sparse randomly-pruned networks may emerge as an attractive option.

**Improving sparse training dynamics**    Many prior works identify various sparse training dynamics issues. In particular, prior works note sparsity impacts weight initialization [35, 31, 49, 11], activation variance [29], gradient flow [61, 37, 57, 11, 1], and step sizes during weight updates [15]. These prior works each only address a subset of these issues in targeted ways, often showing benefits to sparse model training loss. We advocate for a holistic approach, and discuss the relationship between these prior works and our approach in Section 5 after describing and evaluating SµPar.

**Sparse sensitivity to HPs**    Due to the costs of training with fixed weight sparsity, re-using dense HPs is standard practice. Such re-use is typically indicated in appendices or supplemental materials, e.g., [40, 32, 35, 31, 16, 60, 61, 56, 13, 7, 18, 57, 33, 58]. Also, dynamic sparsity approaches often compare to fixed sparsity; these baselines are likewise reported to re-use the dense HPs [2, 41, 10, 34, 11, 59]. However, some prior work has suggested such training is sensitive to HPs, e.g., learning rates [35, 57], learning rate schedules [16], or training length [28], although systematic tuning was not performed. For dynamic sparse training (DST), it is also conventional to re-use dense HPs, whether in dense-to-sparse [37, 15] or sparse-to-sparse (evolving mask) training [2, 8, 34, 11, 59]. As with fixed sparsity, work here has also suggested sensitivity to HPs, e.g., to dropout and label smoothing [16]. DST may also benefit from extra training steps [10] or smaller batch sizes [34], although in DST this may mainly be due to a greater number of opportunities for connectivity exploration [34].

## 3   Sparse maximal update parameterization (SµPar)

We now provide background, motivation, and derivation for SµPar, first introducing notation (Section 3.1) and then defining Feature Learning Desiderata (Section 3.2) with a brief overview of µP (Section 3.3). Finally we motivate SµPar and provide an overview of the parameterization (Section 3.4).

### 3.1   Notation

The operations for a single sparse training step are illustrated in Figure 4. The definition and dimensions are: batch size $B$, learning rate $\eta$, loss function $\mathcal{L}$, forward pass function $\mathcal{F}$, input dimension $d_{\text{in}}$, input activations $\mathbf{X} \in \mathbb{R}^{B \times d_{\text{in}}}$, input activation gradient $\frac{\partial \mathcal{L}}{\partial \mathbf{X}} = \nabla_{\mathbf{X}} \mathcal{L} \in \mathbb{R}^{B \times d_{\text{in}}}$, output dimension $d_{\text{out}}$, output activations $\mathbf{Y} \in \mathbb{R}^{B \times d_{\text{out}}}$, output activation gradient $\frac{\partial \mathcal{L}}{\partial \mathbf{Y}} = \nabla_{\mathbf{Y}} \mathcal{L} \in \mathbb{R}^{B \times d_{\text{out}}}$, weights $\mathbf{W} \in \mathbb{R}^{d_{\text{in}} \times d_{\text{out}}}$, initialization variance $\sigma_W$ for weights $\mathbf{W}$, weight update $\Delta\mathbf{W} \in \mathbb{R}^{d_{\text{in}} \times d_{\text{out}}}$, and $\Delta\mathbf{Y} \in \mathbb{R}^{B \times d_{\text{out}}}$ is the effect of the weight update on output activations: $\Delta\mathbf{Y} = \mathbf{X}(\Delta\mathbf{W} \odot \mathbf{M})$. Unless otherwise specified, $\mathbf{M} \in \{0, 1\}^{d_{\text{in}} \times d_{\text{out}}}$ is an unstructured random static mask with sparsity $s$ and density $\rho = 1 - s$. When changing model scale or sparsity, we refer to a width multiplier $m_d = \frac{d_{\text{in}}}{d_{\text{in, base}}} = \frac{d_{\text{out}}}{d_{\text{out, base}}}$ and density multiplier $m_\rho = \frac{\rho}{\rho_{\text{base}}}$.

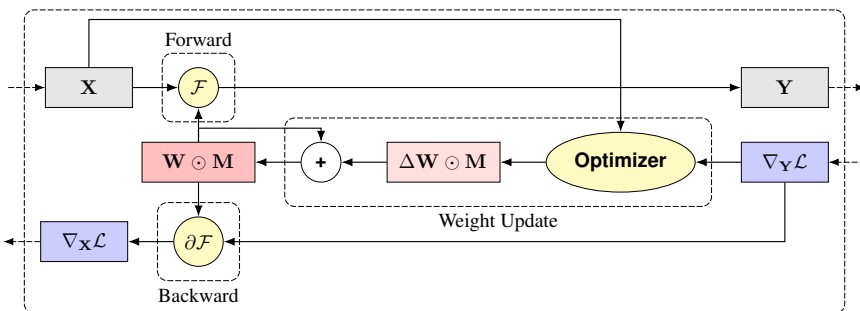

Figure 4: The three operations associated with training a layer with weights that perform the function $\mathcal{F}$: Forward activation calculation, backward gradient propagation, and the weight update.

If we apply sparsity to a linear layer (i.e., $\mathcal{F}$ is a fully-connected layer), our aim is to control:

1. **Forward pass:** $\mathbf{Y} = \mathcal{F}(\mathbf{X}, \mathbf{W} \odot \mathbf{M}) = \mathbf{X}(\mathbf{W} \odot \mathbf{M})$.

2. **Backward pass:** $\nabla_{\mathbf{X}}\mathcal{L} = (\nabla_{\mathbf{Y}}\mathcal{L}) \cdot (\mathbf{W} \odot \mathbf{M})^{\top}$.

3. **Effect of weight update $\Delta\mathbf{W}$ on $\mathbf{Y}$:** $\Delta\mathbf{Y} = \mathbf{X}(\Delta\mathbf{W} \odot \mathbf{M})$[1].

### 3.2 Feature learning: Defining the goal of µP and SµPar

Prior works [64, 63, 65] introduce the Feature Learning Desiderata (FLD) to ensure stable training dynamics as width is varied. Building on prior works, we include gradients $\nabla_{\mathbf{X}}\mathcal{L}$ in the desiderata.

> **Feature Learning Desiderata (FLD):** For layer $l$ and token $i$, we desire that $\|\mathbf{Y}_i^l\|_2 = \Theta(\sqrt{d_{\text{out}}}), \|\nabla_{\mathbf{X}}\mathcal{L}_i^l\|_2 = \Theta(\sqrt{d_{\text{in}}}), \|\Delta\mathbf{Y}_i^l\|_2 = \Theta(\sqrt{d_{\text{out}}}), \forall i, \forall l$.

Recall that if all the entries of some vector $\mathbf{v} \in \mathbb{R}^n$ are some constant $c$, then $\|\mathbf{v}\|_2 = \Theta(\sqrt{n})$ with respect to width $n$. Therefore we can satisfy the FLD by ensuring the *typical element size* of $\mathbf{Y}, \nabla_{\mathbf{X}}\mathcal{L}$, and $\Delta\mathbf{Y}$ is $\Theta(1)$ with respect to **some variable(s)** we would like to scale. Variables to scale include width [64, 63, 65], depth [66, 4], and sparsity (this work). The FLD prescribes a **holistic** signal propagation approach of controlling each of the three operations in a training step, not a subset[2].

### 3.3 Maximal update parameterization (µP)

Here we provide a brief overview of maximal update parameterization (µP) [64, 63, 65]. With the standard parameterization (SP), Yang and Hu [64] show the scale of activations throughout training increases as model width increases, motivating the development of µP. µP [64, 63] is defined as the unique parameterization that satisfies the FLD by ensuring the *typical element size* of $\mathbf{Y}, \nabla_{\mathbf{X}}\mathcal{L}$, and $\Delta\mathbf{Y}$ is $\Theta(1)$ **with respect to change in width** $m_d$. The FLD can also be satisfied by controlling the spectral norm of weights [65]. µP enables µTransfer: the optimum learning rate, initialization weight variance, scalar multipliers, and learning rate schedule all remain consistent as width is increased for µP models [63]. µTransfer can be leveraged to take a *tune small, train large* approach where hyperparameters are extensively tuned for a small model then transferred, enabling reduced tuning budgets and superior tuning for large models compared to standard practice.

### 3.4 Sparse maximal update parameterization (SµPar)

Yang et al. [63] show activation magnitudes explode with increasing model width. In Figure 5 we show sparsity has the opposite effect: increasing sparsity causes shrinking activation magnitudes.

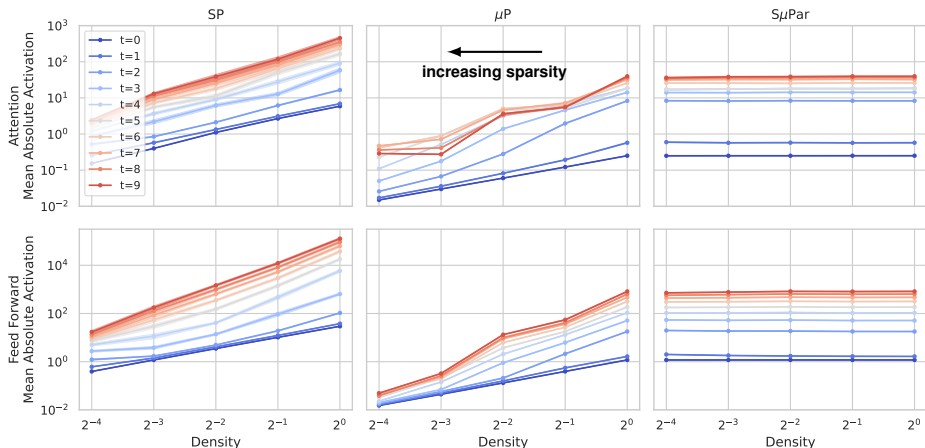

Figure 5: Mean absolute value activations for attention and feed forward blocks after training step $t$ (10 seeds). In SP and µP models, decreasing density causes activations to vanish (note axes on log-scale). In SµPar models, density has little effect on activation scales and there is no vanishing.

---

[1]After a weight update $\Delta\mathbf{W}$ is applied, new output activations can be written as $\mathbf{Y} + \Delta\mathbf{Y} = \mathbf{X}(\mathbf{W} \odot \mathbf{M}) + \mathbf{X}(\Delta\mathbf{W} \odot \mathbf{M})$. Our goal is to control $\Delta\mathbf{Y}$.

[2]For example, initialization methods alone can only control $\|\mathbf{Y}\|_F$ and $\|\nabla_{\mathbf{X}}\mathcal{L}\|_F$ at the first time step.

> **Finding 1**: *Increasing sparsity causes vanishing activations and gradients with both SP and μP.*

SμPar is defined as the unique parameterization that satisfies the FLD by ensuring the *typical element size* of $\mathbf{Y}$, $\nabla_{\mathbf{X}}\mathcal{L}$, and $\Delta\mathbf{Y}$ is $\Theta(1)$ **with respect to change in width $m_d$ and change in density $m_\rho$**. SμPar enables stable activation scales across sparsity levels (Figure 5, right). In this section, we walk through the changes required to control each of the three operations in a sparse training step, providing an overview of the SμPar derivation. We focus on the AdamW [36] optimizer used in our experiments. For a more detailed derivation, including both SGD and Adam, see Appendix D.

**Forward pass at initialization** To ensure the *typical element size* of $\mathbf{Y}$ is $\Theta(1)$ with respect to change in width $m_{d_{\text{in}}}$ and change in density $m_\rho$, we can control the mean and variance of $\mathbf{Y}_{ij}$. Since at initialization $\mathbb{E}[\mathbf{W}] = 0$, $\mathbb{E}[\mathbf{Y}] = 0$, and $\mathbf{W} \perp \mathbf{Y}$, the mean is controlled. The variance of $\mathbf{Y}_{ij}$ can be written as:

$$\text{Var}(\mathbf{Y}_{ij}) = m_{d_{\text{in}}} d_{\text{in,base}} m_\rho \rho_{\text{base}} \sigma_W^2 (\text{Var}(\mathbf{X}) + \mathbb{E}[\mathbf{X}]^2) \tag{1}$$

To ensure $\text{Var}(\mathbf{Y}_{ij})$ scales independent of $m_{d_{\text{in}}}$ and $m_\rho$, we choose $\sigma_{\mathbf{W}}^2 = \frac{\sigma_{\mathbf{W},base}^2}{m_{d_{\text{in}}} m_\rho}$.

**Backward gradient pass at initialization** To ensure the *typical element size* of $\nabla_{\mathbf{X}}\mathcal{L}$ is $\Theta(1)$ with respect to change in width $m_{d_{\text{out}}}$ and change in density $m_\rho$, we can control the mean and variance of $\nabla_{\mathbf{X}}\mathcal{L}$. Since at initialization $\mathbb{E}[\mathbf{W}] = 0$, $\mathbb{E}[\nabla_{\mathbf{X}}\mathcal{L}] = 0$ and the mean is controlled[3]. The variance of $\nabla_{\mathbf{X}}\mathcal{L}_{ij}$ can be written as:

$$\text{Var}(\nabla_{\mathbf{X}}\mathcal{L}_{ij}) = m_{d_{\text{out}}} d_{\text{out,base}} m_\rho \rho_{\text{base}} \sigma_{\mathbf{W}}^2 \text{Var}(\nabla_{\mathbf{Y}}\mathcal{L}) \tag{2}$$

To ensure $\text{Var}(\nabla_{\mathbf{X}}\mathcal{L}_{ij})$ scales independent of $m_{d_{\text{out}}}$ and $m_\rho$, we choose $\sigma_{\mathbf{W}}^2 = \frac{\sigma_{\mathbf{W},base}^2}{m_{d_{\text{out}}} m_\rho}$. Typically $m_{d_{\text{out}}} = m_{d_{\text{in}}}$, allowing the same $\sigma_{\mathbf{W}}^2$ to control both forward and backward scales.

**Effect of Adam weight update $\Delta\mathbf{W}$ on $\mathbf{Y}$** To ensure the *typical element size* of $\Delta\mathbf{Y}$ is $\Theta(1)$ with respect to change in width $m_{d_{\text{out}}}$ and change in density $m_\rho$. By the law of large numbers, the expected size of each element can be written as:

$$\mathbb{E}[\Delta\mathbf{Y}_{ij}] \to \eta m_{d_{\text{in}}} d_{\text{in,base}} m_\rho \rho_{\text{base}} \mathbb{E}\left[\mathbf{X}_{ik}\left(\frac{\sum_t^T \gamma_t \sum_b^B \mathbf{X}_{bk}^t \nabla_{\mathbf{Y}}\mathcal{L}_{bj}^t}{\sqrt{\sum_t^T \omega_t \sum_b^B (\mathbf{X}_{bk}^t \nabla_{\mathbf{Y}}\mathcal{L}_{bj}^t)^2}}\right)\right], \text{ as } (d_{\text{in}}\rho) \to \infty$$

$$\tag{3}$$

To ensure $\Delta\mathbf{Y}_{ij}$ and $\|\Delta\mathbf{Y}\|_F$ are scale invariant to $m_{d_{\text{in}}}, m_\rho$, we choose $\eta = \frac{\eta_{\text{base}}}{m_{d_{\text{in}}} m_\rho}$.

**Implementation summary** Table 1 summarizes the differences between SP, μP, and SμPar. Since we only sparsify hidden weights, SμPar matches μP for input, output, bias, layer-norm, and attention logits. Also note width multipliers $m_d$ and density multipliers $m_\rho$ are usually the same for all layers, allowing simplified notation. This correction is equivalent to μP [63] when $\rho = 1$ and $m_\rho = 1$. The correction to hidden weight initialization we derive is similar to the sparsity-aware initialization in prior work [35, 49, 11]. SμPar should also easily extend to 2:4 sparsity patterns because, in expectation, the rows and columns of $M^l$ should have equal density. A minimal implementation of SμPar is available at https://github.com/EleutherAI/nanoGPT-mup/tree/supar.

## 4 SμPar Training Results

Here, we present empirical results showing the effectiveness of SμPar over SP and μP when training sparse models. When using SP or μP, optimal HPs drift as we change the sparsity level, possibly leading to inconclusive or even reversed findings. SμPar has stable optimal HPs across both model width and sparsity level, and we show it improves over SP and μP across different scaling approaches. Taken together, we see that SμPar sets the Pareto frontier best loss across all sparsities and widths,

---

[3]Although the gradients $\nabla_{\mathbf{Y}}\mathcal{L}$ will have some correlation with weights $\mathbf{W}$ even at initialization, we assume for simplicity that they are fully independent. Future work could investigate this assumption more deeply.

Table 1: Summary of SP, μP, and SμPar implementations.

| Parameterization | SP | μP | SμPar |
|---|---|---|---|
| Embedding Init. Var. | $\sigma_{\text{base}}^2$ | $\sigma_{\text{base}}^2$ | $\sigma_{\text{base}}^2$ |
| Embedding LR | $\eta_{\text{base}}$ | $\eta_{\text{base}}$ | $\eta_{\text{base}}$ |
| Embedding Fwd. | $\mathbf{X}^0 \mathbf{W}_{\text{emb}}$ | $\alpha_{\text{input}} \cdot \mathbf{X}^0 \mathbf{W}_{\text{emb}}$ | $\alpha_{\text{input}} \cdot \mathbf{X}^0 \mathbf{W}_{\text{emb}}$ |
| Hidden Init. Var. | $\sigma_{\text{base}}^2$ | $\sigma_{\text{base}}^2/m_d$ | $\sigma_{\text{base}}^2/(m_d m_\rho)$ |
| Hidden LR (Adam) | $\eta_{\text{base}}$ | $\eta_{\text{base}}/m_d$ | $\eta_{\text{base}}/(m_d m_\rho)$ |
| Unembedding Fwd. | $\mathbf{X}^L \mathbf{W}_{\text{emb}}^\top$ | $\alpha_{\text{output}} \mathbf{X}^L \mathbf{W}_{\text{emb}}^\top/m_d$ | $\alpha_{\text{output}} \mathbf{X}^L \mathbf{W}_{\text{emb}}^\top/m_d$ |
| Attention logits | $\mathbf{Q}^\top \mathbf{K}/\sqrt{d_{\text{head}}}$ | $\mathbf{Q}^\top \mathbf{K}/d_{\text{head}}$ | $\mathbf{Q}^\top \mathbf{K}/d_{\text{head}}$ |

including when we scale to a large dense model with width equal to GPT-3 XL [5]. Optimal *dense* μP HPs—when adjusted using SμPar—are also optimal HPs for all sparse models that we test here.

All tests in this section use GPT-like transformer language models [48, 9], trained on the SlimPajama dataset [54] with a 2048 token context length. We apply random unstructured static sparsity to all projection weights in attention and feedforward blocks while keeping embedding, layer normalization, and bias parameters dense. We refer the reader to Appendix E for full methodology of all experiments.

### 4.1 Sparse hyperparameter transfer

We first show sparsifying a dense model using either SP or μP leads to significant drift in optimal HPs as the sparsity level changes. Figure 6 shows train loss for SP, μP, and SμPar models when trained with varying sparsity levels and sweeping across different peak learning rates. For the SP configuration, as sparsity increases, the optimal learning rate increases in a somewhat unpredictable way. μP experiences similar shift in optimal learning rate, though shifts are even more abrupt. For SμPar, the optimal learning rate is consistently near $2^{-6}$ across all sparsity levels.

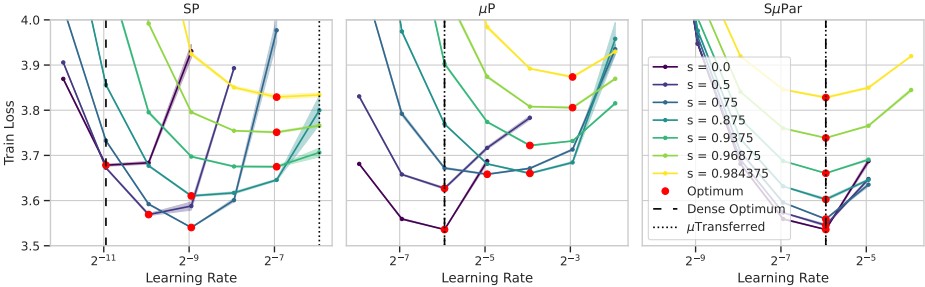

Figure 6: SμPar ensures stable optimal learning rate for any sparsity $s$, unlike SP and μP (3 seeds).

We also sweep base weight initialization values and find even more chaotic behaviors for SP and μP with different sparsity levels (Figure 7, left and center, respectively)[4]. μP even shows discontinuities in optimal initialization values at different sparsity levels. We attribute this discontinuity to widely varying expected activation scales between embedding and transformer decoder layers, where embedding activation scales will tend to dominate for high sparsity levels. SμPar shows consistent optimal initialization (right plot). Figures 6 and 7 demonstrate our second finding.

> **Finding 2**: *With SP and μP, dense and sparse networks do not share the same optimal HPs.*

Figure 8 summarizes our HP transfer tests, showing loss for each parameterization across all sparsities. Even when selecting the best learning rate at each sparsity level for SP and μP, SμPar (largely) forms the Pareto frontier with an average gap of 0.8% better than SP and 2.1% better than μP.

---

[4]These results are taken from a point early in training as models with widely varying initialization tend to become unstable later in training.

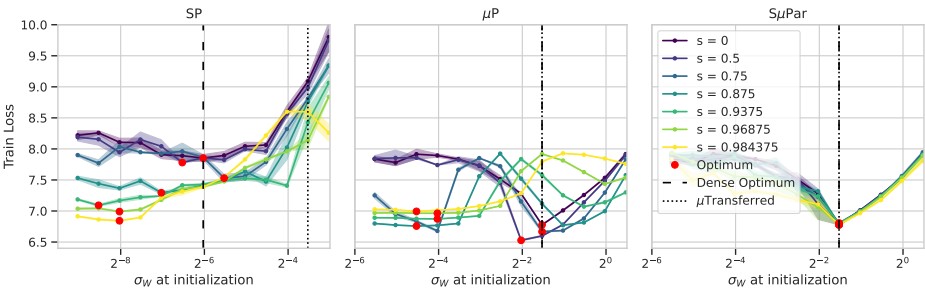

Figure 7: Across sparsity $s$, SP and µP show unstable optimal initialization. SµPar is stable (3 seeds).

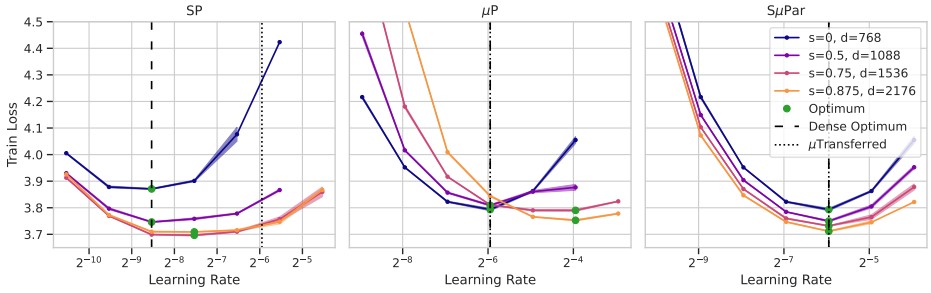

Figure 9: SµPar ensures stable optimal learning rate in Iso-Parameter sparse + wide scaling (3 seeds).

## 4.2 Studying SµPar indicates how some sparse scaling techniques appear to work

So far, we see SµPar can transfer optimal HPs across sparsity levels, but we have also designed it to transfer HPs across different model widths (hidden sizes), similar to µP. Here, we further demonstrate that SµPar transfers optimal HPs across width. More generally, sparse scaling that keeps a fixed number of non-zero weights per neuron allows SP and µP to also transfer HPs.

Figure 9 shows learning rate transfer tests when changing both the model's hidden size, $d_{\mathrm{model}}$, and sparsity level in a common scaling approach called *Iso-Parameter scaling* [18, 10, 59]. Iso-Parameter scaling keeps the model's number of non-zero parameters approximately the same, as width and sparsity are varied[5]. Here, we see the common result that SP models starting from dense HPs *do* tend to

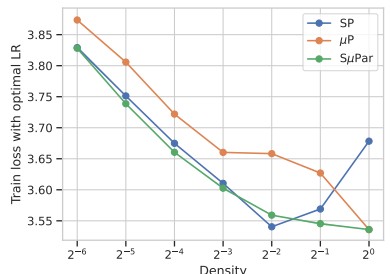

Figure 8: Summarizing loss results from Figure 6 with the optimal learning rate for each parameterization and sparsity.

significantly improve as we increase width and sparsity. Note, though, the optimal learning rate for each sparsity level still shifts. When we correct dense HPs using µP or SµPar, the dense baseline significantly improves, but only SµPar shows consistent loss improvement and stable HPs.

Based on the SµPar formulation: When the number of non-zero weights per neuron (WPN) in the network is the same, µP and SµPar become synonymous, because initialization and learning rate adjustment factors will be constant (i.e., $d_{\mathrm{model}} \cdot \rho = \mathrm{WPN} = O(1)$). Optimized SP HPs will also tend to work well. We define this new scaling setting, which we call Iso-WPN, to verify this hypothesis. In Figure 11, we test SP HPs with Iso-WPN scaling and see the optimal learning rate stays consistently between $2^{-7}$ and $2^{-6}$ with roughly aligned curves (we omit similar µP and SµPar plots for space, because their corrections are the same). The conclusion is that when scaling SP models in an Iso-WPN sparse setting, HPs should maintain similar training dynamics. More generally, as WPN decreases

---

[5]Not perfectly Iso-Parameter due to unsparsified layers (embedding, bias, layer-norm, etc.)

(e.g., by increasing sparsity), the optimal learning rate will tend to increase proportionally, and vice versa[6].

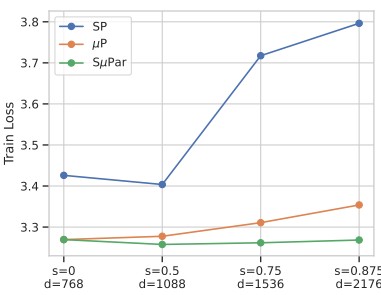

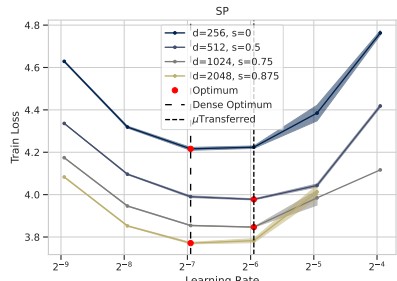

Figure 10: Losses at the end of training when Iso-Parameter scaling.

Figure 11: The SP optimized LR is relatively stable with iso-WPN scaling (3 seeds).

Figures 5, 6, 7, and 9 show SµPar is the only parameterization that ensures stable activation scales and stable optimal HPs across model widths and sparsities, satisfying the FLD.

**Finding 3**: *SµPar enables stable activation and stable optimal HPs for any width and sparsity.*

### 4.3 SµPar scaling to large language model pretraining

We conclude this section reflecting on the demonstration of SµPar improvements in a large-scale language model. We train 610M parameter models starting from a Chinchilla [23] compute-optimal training configuration with 20 tokens per parameter from the SlimPajama dataset. This larger model—with hidden size 2048, 10 layers, and attention head size 64—permits sweeping over a larger range of sparsity levels, so we test up to 99.2% sparsity (density $2^{-7}$).

Figure 3 shows validation loss for each parameterization as we sweep sparsity levels. Additionally, in Table 2, we evaluate the models from Figure 3 on five downstream tasks: ARC-easy, lambada, RACE, PIQA, and BoolQ, which collectively test for common sense reasoning, world knowledge, and reading comprehension. As sparsity increases, results across pretraining loss and average downstream task accuracy consistently show SP and µP fall farther behind SµPar. Since these models are trained with a large number of tokens, we attribute the widening loss gap mostly to increasingly under-tuned learning rates for SP and µP as sparsity increases–the weight updates lose gradient information throughout training. Figure 8 shows retuning SP and µP could recover some of the gap to SµPar, but that could be costly: These runs take 3-6 hours on a Cerebras CS-3 system (or > 9 days on an NVIDIA A100 GPU).

Finally, returning to the Iso-Parameter scaling setting, Figure 10 shows losses for 111M parameter models trained on 1B tokens and scaled up while using dense optimal HPs. The SP and µP models experience detuning as sparsity increases, allowing SµPar to achieve superior losses[7].

**Finding 4**: *Sparse networks trained with SµPar improve over SP and µP due to improved tuning.*

### 4.4 Dynamic sparsity hyperparameter transfer

In Figure 12 we test the transfer of optimal learning rate across sparsity levels for two popular dynamic sparse training methods: Rigging the Lottery (RigL) [10][8] and Gradual Magnitude Pruning (GMP) [68][9]. We show that none of SP, µP, or SµPar achieve transfer of optimal learning rate across sparsity levels. For SP and µP we see that higher sparsity levels have higher optimal learning rates.

---

[6]Our results generalize the Yang et al. finding that optimal LR decreases as width increases [63, Figure 1].

[7]Note this is not an Iso-FLOP comparison because increasing $d_{\text{model}}$ also increases attention dot product and embedding FLOPs, which aren't be sparsified. This is so significant that our 87.5% sparse model from Figure 10 has double the training FLOPs of the dense baseline, with virtually unchanged loss.

[8]RigL: Uniform sparsity distribution, drop fraction of 0.3, and mask updates every 100 steps.

[9]GMP: Uniform sparsity distribution, cubic sparsity schedule, and mask updates every 100 steps.

This is because sparsity reduces activation and gradient scales such that a larger learning rate is needed to counteract this. SµPar sees the opposite trend where higher sparsity levels have lower optimal learning rates, indicating that SµPar is "overcorrecting".

Dynamic sparse methods can make updates to the weight mask such that the distribution of unmasked/non-zero weights changes to something non-Gaussian, which prevents SµPar from being correct in expectation. Compared to random pruning, a mask obtained from magnitude pruning will better preserve the size of activations and gradients seen in the dense network. Since SµPar assumes weights are drawn from a Gaussian distribution, SµPar ends up "overcorrecting" the initialization and learning rate. In future work it would be impactful to develop a parameterization which generalizes SµPar to work for an arbitrary sparse training algorithm.

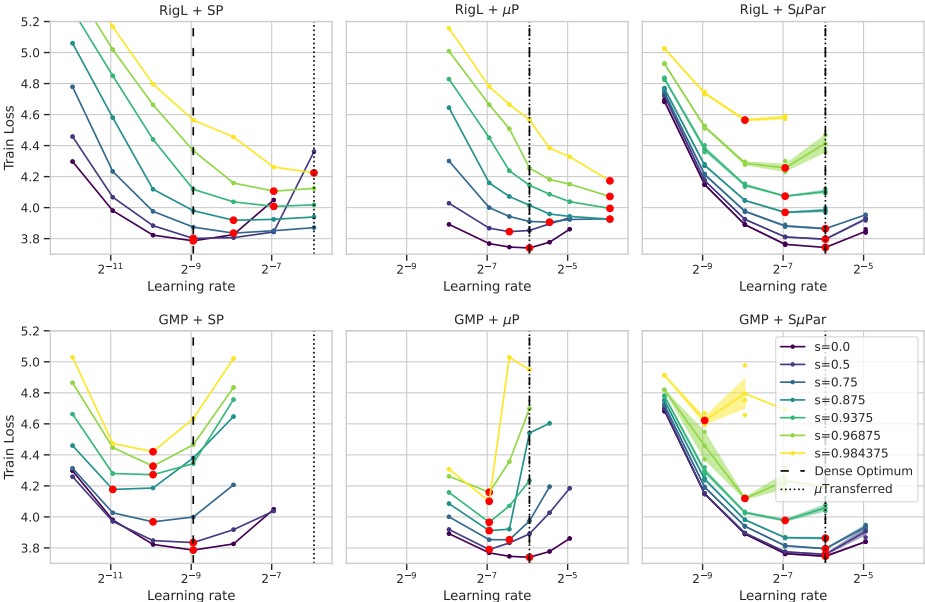

Figure 12: For dynamic sparse training methods RigL and GMP, none of SP, µP, or SµPar achieve stable optimal learning rate across sparsity (3 seeds). Missing points indicate diverged training runs.

## 5 Discussion

To improve sparse training, prior works make targeted corrections which arise from observations that sparsity can cause degraded activation, gradient, and/or weight update signal propagation. We review these observations and corrections to advocate for holistic control of sparse training dynamics.

**Sparsifying Can Cause Vanishing Activations**   Evci et al. [11] note that by initializing weights using dense methods (e.g., [17, 21]), the "vast majority" of sparse networks have vanishing activations. Lasby et al. [29, App. A] analyze activation variance as a guide for selecting structured sparsity. The FLD suggest activation norms be measured and controlled with respect to sparsity, so activation variance can be considered a proxy to whether sparsity might negatively impact training dynamics. Evci et al. [11] ultimately initialize variances via neuron-specific sparse connectivity, while Liu et al. [35] and Ramanujan et al. [49] propose scaling weight variances proportional to layer sparsity. These corrections, however, only target controlling activations but not weight updates.

**Gradient Flow Partially Measures the Weight Update µDesideratum**   Sparsity also impairs *gradient flow*—the magnitude of the gradient to the weights—during training [11, 1]. Since gradient flow is measured using the norm of the weight gradients, it measures a piece of the weight update. Unfortunately, gradient flow does not directly measure the effect of the weight update step, which can also involve adjustments for things like optimizer state (e.g., momentum and velocity), the learning rate, and weight decay. Prior works propose techniques to improve gradient flow during sparse

training and pruning by adjusting individual hyperparameters or adding normalization [61, 37, 11, 1]. However, these techniques might overlook the effects of the optimizer and learning rates in weight updates. Notably, Tessera et al. [57] *do* consider some of these effects, but their proposed techniques maintain gradient flow only in the Iso-Parameter scaling setting rather than arbitrary sparsification.

Frantar et al. [15, App. A.1] also endeavor to control weight updates, where they observe diminished step sizes when optimizing sparse networks with Adafactor [52]. They correct this by computing Adafactor's root-mean-square scaling adjustments over *unpruned* weights and updates. However, such normalization does not prevent activations from scaling with model width [63, 65]. In this sense, sparsity-aware fixes to Adafactor can improve dynamics, but will not address instability holistically. In Figure 14 we show the SμPar LR correction alone is not even sufficient to achieve stable optimal $\eta$.

**Weight Initialization Only Controls Dynamics at Initialization**   We noted works above that adjust sparse weight initializations [11, 35, 49]. Additionally, Lee et al. [31] explore orthogonal weight initialization [46], both before pruning (to ensure SNIP [32] pruning scores are on a similar scale across layers) and after (to improve trainability of the sparse network). While adjusting weights can improve sparse training dynamics at initialization, such adjustments are insufficient to stabilize signals *after multiple steps of training*, in the same way that standard weight initializations fail to stabilize training of dense networks. In Figure 13 we show the SμPar init. alone is not even sufficient to achieve stable optimal $\sigma_W$.

# 6   Limitations

As Section 4.4 shows, SμPar requires further extension for dynamic sparse training due to unpredictable changes in weight distributions. The same applies to methods which prune at initialization or after pretraining in a non-random fashion. Iterative magnitude pruning (IMP) is an interesting case since it involves rewinding weights back to their initial values while maintaining the same mask [12]. If the IMP mask at initialization still allows the non-zero weights to have a Gaussian distribution, then SμPar would apply to this case. Therefore, it's possible SμPar *could* prevent "HP detuning" in later IMP iterations, and *potentially* improve IMP losses, though we do not explore this. SμPar would also work for random structured pruning of entire neurons at initialization because this case simply reduces to training with a narrower dense model.

For weight sparsity more generally, the most pressing limitation is the lack of hardware acceleration [38]. While new software [50, 29, 43] continues to better leverage existing hardware, the growth of software and hardware co-design is also encouraging [59, 6], as effective sparsity techniques can be specifically optimized in deep learning hardware. But to effectively plan hardware, we need to train and test sparse prototypes at next-level sizes, at scales where the optimum sparsity level may be higher than in current networks [15]. Performing such *scaling law*-style studies requires incredible resources even for dense models with well-established training recipes [27, 23]. As SμPar reduces training and tuning costs, it can help unlock these studies and guide future hardware design.

Finally, the scaling factors for the weight update need to be derived for each optimizer, which might limit the usability of SμPar in practice. For a discussion of the broader impacts of SμPar, see Appendix A.

# 7   Conclusion

Nobody said training with sparsity was easy. We showed that with the standard parameterization and μP, increasing sparsity level directly correlates with vanishing activations. Impaired training dynamics prevent sparse models from sharing the same optimal hyperparameters, suggesting prior results based on re-use of dense HPs merit re-examination. In contrast, by holistically controlling the training process, SμPar prevents vanishing activations and enables HP transfer (across both width and sparsity). LLMs trained with SμPar improve over μP and the standard parameterization. As such, we hope SμPar makes things a little easier for sparsity research going forward.

## Acknowledgements

We would like to thank Gavia Gray, who provided helpful feedback on the manuscript, and Gurpreet Gosal, who tuned the µTransferred hyperparameters seen throughout the document.

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

## A  Broader impacts

Sparsity is recognized to reduce carbon emissions [45] and offset well-known environmental and financial costs of large model training [3]. For example, unstructured sparsity can be accelerated by the Cerebras Wafer-Scale Engine[10] and 2:4 block sparsity can be accelerated by NVIDIA Ampere GPUs[11]. There is growing recognition that HP tuning is a key contributor to these costs. HP tuning is costly, possibly undermining equity in AI research due to financial resources [55]. During model *re*training, *sensitivity* to HPs also leads to downstream costs [55]. SμPar can reduce these costs and sensitivities and thus improve equity.

Sparsity also has potential drawbacks. Hooker et al. [24] showed that even when top-line performance metrics are comparable, pruned networks may perform worse on specific subsets of the data (including on underrepresented groups [25]), may amplify sensitivity to adversarial examples, and may be more sensitive to distribution shift. These sensitivities may depend on the degree of sparsity [20]. It remains an open question whether such drawbacks occur only with pruning or when training with sparsity from scratch (as in SμPar) [22], and how such sensitivity may impact susceptibility to misuse [62]. We require sparsity-specific methods to detect [53, 42] and mitigate [19, 44] harm. Moreover, since many large models are later pruned for deployment, we recommend testing and documenting in the model card [39] any adverse affects of sparsification at the time of model release.

## B  Downstream task comparison of parameterizations

In Table 2, we evaluate the models from Figure 3 on five downstream tasks: ARC-easy, lambada, RACE, PIQA, and BoolQ, which collectively test for common sense reasoning, world knowledge, and reading comprehension. We also specifically chose tasks that are easy enough for even extremely sparse models to significantly outperform random chance.

| Sparsity | - | 0 | | | 0.5 | | | 0.75 | | | 0.875 | | |
|---|---|---|---|---|---|---|---|---|---|---|---|---|---|
| | Rand. | SP | μP | SμPar | SP | μP | SμPar | SP | μP | SμPar | SP | μP | SμPar |
| ARC-easy | 0.25 | 0.49 | **0.51** | **0.51** | 0.45 | **0.49** | 0.48 | 0.44 | **0.45** | **0.45** | 0.42 | 0.43 | **0.44** |
| LAMBADA | 0.00 | 0.32 | **0.36** | **0.36** | 0.27 | 0.31 | **0.32** | 0.22 | 0.27 | **0.28** | 0.20 | 0.23 | **0.25** |
| RACE | 0.25 | **0.30** | **0.30** | **0.30** | 0.29 | **0.31** | 0.30 | 0.28 | **0.30** | 0.29 | 0.27 | **0.28** | **0.28** |
| PIQA | 0.50 | **0.67** | **0.67** | **0.67** | 0.63 | 0.65 | **0.67** | 0.63 | **0.64** | **0.64** | 0.62 | **0.63** | **0.63** |
| BoolQ | 0.50 | 0.53 | **0.57** | **0.57** | **0.58** | 0.55 | 0.51 | 0.57 | **0.62** | 0.52 | 0.61 | **0.62** | **0.62** |
| Avg. | 0.30 | 0.46 | **0.48** | **0.48** | 0.44 | **0.46** | **0.46** | 0.43 | **0.46** | 0.44 | 0.42 | **0.44** | **0.44** |
| Sparsity | - | 0.9375 | | | 0.96875 | | | 0.984375 | | | 0.992188 | | |
| | Rand. | SP | μP | SμPar | SP | μP | SμPar | SP | μP | SμPar | SP | μP | SμPar |
| ARC-easy | 0.25 | 0.41 | 0.40 | **0.43** | 0.39 | **0.42** | 0.41 | 0.38 | 0.38 | **0.41** | 0.37 | **0.38** | 0.38 |
| LAMBADA | 0.00 | 0.19 | 0.20 | **0.21** | 0.16 | 0.18 | **0.19** | 0.13 | 0.15 | **0.17** | 0.12 | 0.13 | **0.14** |
| RACE | 0.25 | 0.25 | 0.27 | **0.28** | 0.24 | 0.25 | **0.27** | 0.24 | 0.24 | **0.25** | 0.25 | 0.24 | **0.26** |
| PIQA | 0.50 | **0.61** | **0.61** | **0.61** | 0.60 | **0.61** | 0.60 | 0.59 | **0.60** | 0.59 | 0.58 | **0.60** | **0.60** |
| BoolQ | 0.50 | **0.62** | **0.62** | 0.61 | 0.57 | **0.62** | **0.62** | 0.45 | **0.62** | 0.61 | 0.41 | **0.61** | **0.61** |
| Avg. | 0.30 | 0.42 | 0.42 | **0.43** | 0.39 | **0.42** | **0.42** | 0.36 | 0.40 | **0.41** | 0.34 | 0.39 | **0.40** |

Table 2: Downstream evaluation accuracy; higher is better: SμPar performs best or within 0.01 of best across all sparsity levels and tasks, except *boolq* at 50% and 75% sparsity. Even at 99% sparsity, SμPar models maintain 40%+ average accuracy, whereas the SP model drops to 34%, close to the 30% accuracy of the random baseline.

## C  Individual ablations of SμPar initialization and learning rate corrections

In Figures 13 and 14, we individually ablate the effect of the SμPar initialization and the SμPar learning rate. We show that using only the SμPar initialization in conjunction with μP (μP + SμPar initialization only) does not allow for transfer of optimal initialization standard deviation or optimal learning rate across sparsity levels. We also show that using only the SμPar learning rate in conjunction with μP does not achieve transfer either. Therefore, **both** the SμPar initialization and learning rate corrections are required to achieve optimal hyperparameter transfer across sparsity levels.

---

[10] https://www.cerebras.net/blog/harnessing-the-power-of-sparsity-for-large-gpt-ai-models
[11] https://www.nvidia.com/en-us/data-center/ampere-architecture/

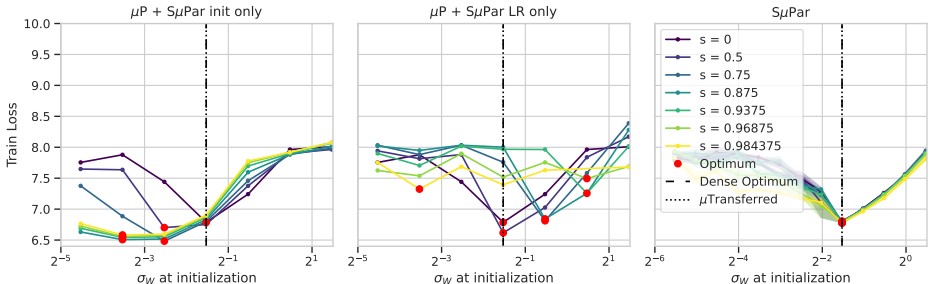

Figure 13: SμPar ensures stable optimal weight initialization standard deviation, unlike SP, μP, μP + SμPar init only, and μP + SμPar LR only.

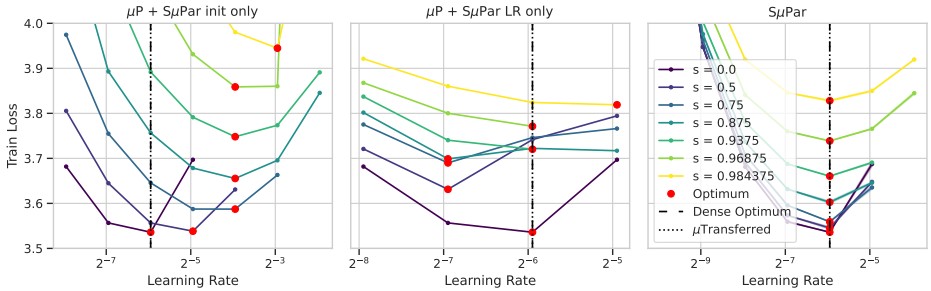

Figure 14: SμPar ensures stable optimal learning rate (**Bottom**), unlike SP, μP, μP + SμPar init only, and μP + SμPar LR only.

# D  SμPar detailed derivation

## D.1  Forward pass at initialization

The first stage where we would like to control training dynamics is in the layer's forward function. For a random unstructured sparsity mask $\mathbf{M}$, since each *column* of $\mathbf{M}$ has $d_{\text{in}}\rho$ non-zero elements in expectation, we can rewrite the forward pass as:

$$\mathbf{Y}_{ij} = [\mathbf{X}(\mathbf{W} \odot \mathbf{M})]_{ij} = \sum_{q=1}^{d_{\text{in}}} \mathbf{X}_{iq}(\mathbf{W}_{qj} \cdot \mathbf{M}_{qj}) = \sum_{k:\mathbf{M}_{kj}=1}^{d_{\text{in}}\rho} \mathbf{X}_{ik}\mathbf{W}_{kj} \qquad (4)$$

To satisfy the FLD, we desire the *typical element size* of $\mathbf{Y}$ is $\Theta(1)$ with respect to change in width $m_{d_{\text{in}}}$ and change in density $m_\rho$. To achieve this we can ensure the mean and variance of $\mathbf{Y}_{ij}$ are invariant to $m_{d_{\text{in}}}$ and $m_\rho$.

**Mean:** As expectation is linear and $\mathbf{X}$ and $\mathbf{W}$ are independent at initialization:

$$\mathbb{E}[\mathbf{Y}_{ij}] = \mathbb{E}\left[\sum_{k:\mathbf{M}_{kj}=1}^{d_{\text{in}}\rho} \mathbf{Y}_{ik}\mathbf{W}_{kj}\right] = \sum_{k:\mathbf{M}_{kj}=1}^{d_{\text{in}}\rho} \mathbb{E}[\mathbf{X}_{ik}\mathbf{W}_{kj}] = \sum_{k:\mathbf{M}_{kj}=1}^{d_{\text{in}}\rho} \mathbb{E}[\mathbf{X}_{ik}]\mathbb{E}[\mathbf{W}_{kj}] \qquad (5)$$

Therefore, since at initialization $\mathbb{E}[\mathbf{W}_{ij}] = 0$, $\mathbb{E}[\mathbf{Y}_{ij}] = 0$ and the mean is controlled.

**Variance:** As expectation is linear and each weight element is IID:

$$\text{Var}(\mathbf{Y}_{ij}) = \text{Var}\left(\sum_{k:\mathbf{M}_{kj}=1}^{d_{\text{in}}\rho} \mathbf{X}_{ik}\mathbf{W}_{kj}\right) = \sum_{k:\mathbf{M}_{kj}=1}^{d_{\text{in}}\rho} \text{Var}(\mathbf{X}_{ik}\mathbf{W}_{kj}) \qquad (6)$$

Then, since $\mathbf{X}$ and $\mathbf{W}$ are independent at initialization:

$$\text{Var}(\mathbf{Y}_{ij}) = \sum_{k:\mathbf{M}_{kj}=1}^{d_{\text{in}}\rho} (\text{Var}(\mathbf{X}_{ik}) + \mathbb{E}[\mathbf{X}_{ik}]^2)(\text{Var}(\mathbf{W}_{kj}) + \mathbb{E}[\mathbf{W}_{kj}]^2) - (\mathbb{E}[\mathbf{X}_{ik}]\mathbb{E}[\mathbf{W}_{kj}])^2 \quad (7)$$

Finally, since at initialization $\mathbb{E}[\mathbf{W}_{kj}] = 0$ and redefining $\text{Var}(\mathbf{W}_{kj}) = \sigma_\mathbf{W}^2$:

$$\text{Var}(\mathbf{Y}_{ij}) = \sum_{k:\mathbf{M}_{kj}=1}^{d_{\text{in}}\rho} (\text{Var}(\mathbf{X}_{ik}) + \mathbb{E}[\mathbf{X}_{ik}]^2)\text{Var}(\mathbf{W}_{kj}) = d_{\text{in}}\rho\sigma_\mathbf{W}^2(\text{Var}(\mathbf{X}) + \mathbb{E}[\mathbf{X}]^2) \quad (8)$$

Rewriting in terms of multipliers for the width $m_{d_{\text{in}}} = \frac{d_{\text{in}}}{d_{\text{in, base}}}$ and the change in density $m_\rho = \frac{\rho}{\rho_{\text{base}}}$:

$$\text{Var}(\mathbf{Y}_{ij}) = m_{d_{\text{in}}}d_{\text{in, base}}m_\rho\rho_{\text{base}}\sigma_\mathbf{W}^2(\text{Var}(\mathbf{X}) + \mathbb{E}[\mathbf{X}]^2) \quad (9)$$

**Solution:** To satisfy the FLD and ensure $\text{Var}(\mathbf{Y}_{ij})$ scales independently of $m_{d_{\text{in}}}$ and $m_\rho$, we choose to set $\sigma_\mathbf{W}^2 = \frac{\sigma_{\mathbf{W},base}^2}{m_{d_{\text{in}}}m_\rho}$. This ensures typical entry size of $\mathbf{Y}$ is invariant to changes in width $m_{d_{\text{in}}}$ and density $m_\rho$.

Note that this correction is equivalent to µP [63] when $m_\rho = 1$. Further, the sparsity factor in the denominator matches the correction for sparsity-aware initialization from Evci et al. [11].

### D.2  Backward gradient pass at initialization

The next stage we would like to control training dynamics is in the layer's backward pass. For a random unstructured sparsity mask $\mathbf{M}$, since each *row* of $\mathbf{M}$ has $d_{\text{out}}\rho$ non-zero elements in expectation, we can rewrite the backward pass as:

$$\nabla_\mathbf{X}\mathcal{L}_{ij} = \left[\nabla_\mathbf{Y}\mathcal{L}(\mathbf{W}\odot\mathbf{M})^\top\right]_{ij} = \sum_q^{d_{\text{out}}} \nabla_\mathbf{Y}\mathcal{L}_{iq}(\mathbf{W}_{jq}\cdot\mathbf{M}_{jq}) = \sum_{k:\mathbf{M}_{jk}=1}^{d_{\text{out}}\rho} \nabla_\mathbf{Y}\mathcal{L}_{ik}\mathbf{W}_{jk} \quad (10)$$

To satisfy the FLD, we desire the *typical element size* of $\nabla_\mathbf{X}\mathcal{L}$ is $\Theta(1)$ with respect to change in width $m_{d_{\text{out}}}$ and change in density $m_\rho$. To achieve this, we can ensure the mean and variance of $\nabla_\mathbf{X}\mathcal{L}$ are invariant to $m_{d_{\text{out}}}$ and $m_\rho$.

**Mean:** Although the gradients $\nabla_\mathbf{Y}\mathcal{L}$ will have some correlation with weights $\mathbf{W}$ even at initialization, we assume for simplicity that they are fully independent. Future work could investigate this assumption more deeply. As expectation is linear:

$$\mathbb{E}[\nabla_\mathbf{X}\mathcal{L}_{ij}] = \mathbb{E}\left[\sum_{k:\mathbf{M}_{jk}=1}^{d_{\text{out}}\rho} \nabla_\mathbf{Y}\mathcal{L}_{ik}\mathbf{W}_{jk}\right] = \sum_{k:\mathbf{M}_{jk}=1}^{d_{\text{out}}\rho} \mathbb{E}[\nabla_\mathbf{Y}\mathcal{L}_{ik}\mathbf{W}_{jk}] = \sum_{k:\mathbf{M}_{jk}=1}^{d_{\text{out}}\rho} \mathbb{E}[\nabla_\mathbf{Y}\mathcal{L}_{ik}]\mathbb{E}[\mathbf{W}_{jk}]$$

$$(11)$$

Therefore, since at initialization $\mathbb{E}[\mathbf{W}_{jk}] = 0$, $\mathbb{E}[\nabla_\mathbf{X}\mathcal{L}_{ij}] = 0$, the mean is controlled.

**Variance:** As expectation is linear and each weight element is IID:

$$\text{Var}(\nabla_\mathbf{X}\mathcal{L}_{ij}) = \text{Var}\left(\sum_{k:\mathbf{M}_{jk}=1}^{d_{\text{out}}\rho} \nabla_\mathbf{Y}\mathcal{L}_{ik}\mathbf{W}_{jk}\right) = \sum_{k:\mathbf{M}_{jk}=1}^{d_{\text{out}}\rho} \text{Var}(\nabla_\mathbf{Y}\mathcal{L}_{ik}\mathbf{W}_{jk}) \quad (12)$$

From the backward pass mean derivation, we know $\mathbb{E}[\nabla_\mathbf{Y}\mathcal{L}_{ij}] = 0$. Then, similar to the forward pass variance derivation, we can simplify using the facts that at initialization, $\nabla_\mathbf{Y}\mathcal{L}$ and $\mathbf{W}$ are (roughly) independent and $\mathbb{E}[\mathbf{W}] = 0$. Similarly we can also define $\text{Var}(\mathbf{W}_{kj}^l) = \sigma_\mathbf{W}^2$ and rewrite in terms of width multiplier $m_{d_{\text{out}}} = \frac{d_{\text{out}}}{d_{\text{out,base}}}$ and changes in density $m_\rho = \frac{\rho}{\rho_{\text{base}}}$:

$$\text{Var}(\nabla_\mathbf{X}\mathcal{L}_{ij}) = m_{d_{\text{out}}}d_{\text{out,base}}m_\rho\rho_{\text{base}}\sigma_\mathbf{W}^2\text{Var}(\nabla_\mathbf{Y}\mathcal{L}) \quad (13)$$

**Solution:** To satisfy the FLD and ensure $\text{Var}(\nabla_\mathbf{X}\mathcal{L}_{ij})$ scales independently of $m_{d_{\text{out}}}$ and $m_\rho$, we choose to set $\sigma_\mathbf{W}^2 = \frac{\sigma_{\mathbf{W},\text{base}}^2}{m_{d_{\text{out}}}m_\rho}$. This ensures the typical entry size of $\nabla_\mathbf{X}\mathcal{L}$ is invariant to changes

in width $m_{d_{\text{out}}}$ and density $m_\rho$. Typically, we scale model width such that $m_{d_{\text{out}}} = m_{d_{\text{in}}}$. This equal scaling allows the same initialization variance to correct both forward activation and backward gradient scales, making them independent of width. Further, since we assume random sparsity across layer's weights, the sparsity initialization correction factor, $m_\rho$, is the same for both the forward activations and backward gradients.

### D.3  Effect of weight update $\Delta \mathbf{W}$ on $\mathbf{Y}$

To satisfy the FLD, we desire the *typical element size* of the weight update $\Delta \mathbf{Y}$ is $\Theta(1)$ with respect to change in width $m_{d_{\text{in}}}$ and change in density $m_\rho$. To achieve this we examine the expected size of each element. Here, we use $\eta$ to be the learning rate for this layer. For a random unstructured sparsity mask $\mathbf{M}$, since each *column* of $\mathbf{M}$ has $d_{\text{in}}\rho$ non-zero elements in expectation:

$$\Delta \mathbf{Y}_{ij} = [\eta \mathbf{X}(\Delta \mathbf{W} \odot \mathbf{M})]_{ij} = \eta \sum_{q=1}^{d_{\text{in}}} \mathbf{X}_{iq}(\Delta \mathbf{W}_{qj} \cdot \mathbf{M}_{qj}) = \eta \sum_{k:\mathbf{M}_{kj}=1}^{d_{\text{in}}\rho} \mathbf{X}_{ik}\Delta \mathbf{W}_{kj} \quad (14)$$

**Mean:** As expectation in linear:

$$\mathbb{E}[\Delta \mathbf{Y}_{ij}] = \mathbb{E}\left[\eta \sum_{k:\mathbf{M}_{kj}=1}^{d_{\text{in}}\rho} \mathbf{X}_{ik}\Delta \mathbf{W}_{kj}\right] = \eta \sum_{k:\mathbf{M}_{kj}=1}^{d_{\text{in}}\rho} \mathbb{E}[\mathbf{X}_{ik}\Delta \mathbf{W}_{kj}] \quad (15)$$

Since $\Delta \mathbf{W}$ was derived from $\mathbf{X}$, there is covariance between these variables and $\mathbb{E}[\mathbf{X}_{ik}\Delta \mathbf{W}_{kj}]$ is non-zero. By the Law of Large Numbers:

$$\mathbb{E}[\Delta \mathbf{Y}_{ij}] \to \eta d_{\text{in}}\rho \mathbb{E}[\mathbf{X}_{ik}\Delta \mathbf{W}], \text{ as } (d_{\text{in}}\rho) \to \infty \quad (16)$$

Rewriting in terms of width and density multipliers:

$$\mathbb{E}[\Delta \mathbf{Y}_{ij}] \to \eta m_{d_{\text{in}}} d_{\text{in,base}} m_\rho \rho_{\text{base}} \mathbb{E}[\mathbf{X}_{ik}\Delta \mathbf{W}], \text{ as } (d_{\text{in}}\rho) \to \infty \quad (17)$$

Equation 17 will be used as intermediate result in the following sections.

#### D.3.1  Effect SGD weight update $\Delta \mathbf{W}$ on $\mathbf{Y}$

Following the formulation in [63], stochastic gradient descent (SGD) weight updates take the form:

$$\Delta \mathbf{W}_{kj}^l = \left[\frac{(\mathbf{X})^\top (\nabla_{\mathbf{Y}}\mathcal{L})}{d_{\text{in}}}\right]_{kj} = \frac{1}{d_{\text{in}}} \sum_{b=1}^{B} \mathbf{X}_{bk}(\nabla_{\mathbf{Y}}\mathcal{L})_{bj} \quad (18)$$

so we can rewrite Equation 17 as:

$$\mathbb{E}[\Delta \mathbf{Y}_{ij}] \to \eta m_\rho \rho_{\text{base}} \mathbb{E}\left[\mathbf{X}_{ik} \sum_{b=1}^{B} \mathbf{X}_{bk}(\nabla_{\mathbf{Y}}\mathcal{L})_{bj}\right], \text{ as } (d_{\text{in}}\rho) \to \infty \quad (19)$$

**Solution:** For SGD updates, to satisfy the FLD and ensure $\mathbb{E}[\Delta \mathbf{Y}_{ij}]$ and the typical entry size of $\Delta \mathbf{Y}$ are scale invariant to $m_d$ and $m_\rho$, we choose $\eta = \eta_{\text{base}}/m_\rho$. Note this correction is equivalent to µP [63] when $\rho = 1, m_\rho = 1$.

#### D.3.2  Effect of Adam weight update $\Delta \mathbf{W}$ on $\mathbf{Y}$

Following the formulation in Yang et al. [63], Adam weight updates take the form:

$$\Delta \mathbf{W}_{kj} = \frac{\sum_t^T \gamma_t \sum_b^B \mathbf{X}_{bk}^{l,t}(\nabla_{\mathbf{Y}}\mathcal{L})_{bj}^t}{\sqrt{\sum_t^T \omega_t \sum_b^B (\mathbf{X}_{bk}^t(\nabla_{\mathbf{Y}}\mathcal{L})_{bj}^t)^2}} \quad (20)$$

where $T$ is the current training step and $\gamma_t, \omega_t$ are the moving average weights at each training step. Here, we can just consider the weight update associated with an unpruned weight, since a pruned

weight will have value and update 0 (i.e., pruned weights trivially satisfy that their effect on forward activations cannot depend on width or sparsity). We can rewrite Equation 17 as:

$$\mathbb{E}[\Delta\mathbf{Y}_{ij}] \to \eta m_{d_{\text{in}}} d_{\text{in,base}} m_\rho \rho_{\text{base}} \mathbb{E}\left[\mathbf{X}_{ik}\left(\frac{\sum_t^T \gamma_t \sum_b^B \mathbf{X}_{bk}^t \nabla_{\mathbf{Y}} \mathcal{L}_{bj}^t}{\sqrt{\sum_t^T \omega_t \sum_b^B (\mathbf{X}_{bk}^t \nabla_{\mathbf{Y}} \mathcal{L}_{bj}^t)^2}}\right)\right], \text{ as } (d_{\text{in}}\rho) \to \infty \tag{21}$$

**Solution:** For Adam updates, to satisfy the FLD and ensure $\mathbb{E}[\Delta\mathbf{Y}_{ij}]$ and the typical entry size of $\Delta\mathbf{Y}$ are scale invariant to $m_{d_{\text{in}}}$ and $m_\rho$, we choose $\eta = \frac{\eta_{\text{base}}}{m_{d_{\text{in}}} m_\rho}$. Note that this correction is equivalent to μP [63] when $\rho = 1, m_\rho = 1$.

### D.4 Additional notes about derivation

We make a few supplementary notes about the above derivation:

- Throughout our derivation, we use $\rho$ to refer to the density level. Note that since this derivation is local to a single layer in the model, the density (or sparsity) level can also be parameterized independently for each layer. If a sparsity technique will use layer-wise independent sparsity levels, appropriate corrections should be made for each layer.

- Similar to the $\rho$ notation, we use $\eta$ to denote the learning rate, but this learning rate can be layer-specific depending on sparsity level. Appropriate corrections must be made if using layer-wise independent sparsities.

- The use of the Law of Large Numbers in portions of the above derivation indicate that SμPar is expected to provide stable training dynamics as the number of non-zero weights per neuron (WPN) tends to infinity. However, in sparse settings, the WPN can tend to be small. If WPN is small, training dynamics may be affected, and this might be a direction for future work.

- In this work, we only consider sparsifying linear projection layers. As a result, SμPar matches μP for other layers like input, output, bias, layer-norm, and attention logits. Depending on the sparsification technique, these other layers might need to be reviewed for their effects on training dynamics.

## E   Experimental details

**SμPar base hyperparameter tuning**   To find the optimized set of hyperparameters for SμPar, we actually tune μP HPs on a dense proxy model. By formulation of SμPar, these HPs transfer optimally to all the sparse models trained for this work. This dense proxy model is a GPT-2 model, but with small changes: ALiBi position embeddings [47] and SwiGLU nonlinearity [51]. We configure it with width: $d_{\text{model}} = d_{\text{model,base}} = 256$, number of layers: $n_{\text{layers}} = 24$, and head size: $d_{\text{head}} = 64$, resulting in 39M parameters. We trained this proxy model on 800M tokens with a batch size of 256 sequences and sequence length 2048 tokens. We randomly sampled 350 configurations of base learning rates, base initialization standard deviation, and embedding and output logits scaling factors. From this sweep we obtained the tuned hyperparameters listed in Table 3.

Table 3: Tuned hyperparameters for our dense proxy model.

| Hyperparameter | Value |
|---|---|
| $\sigma_{W,\text{base}}$ | 0.08665602 |
| $\eta_{\text{base}}$ | 1.62E-2 |
| $\alpha_{\text{input}}$ | 9.1705 |
| $\alpha_{\text{output}}$ | 1.0951835 |

**Experimental details for all figures**   In Table 4, we provide extensive details on hyperparameters, model size, and training schedule for all experiments in this paper. All models in this paper were trained on the SlimPajama dataset [54], a cleaned and deduplicated version of the RedPajama dataset.

Table 4: Experimental details for all figures in this paper.

| Figure | $d_{model}$ | $L$ | $d_{head}$ | $B$ | LR | Init. Stdev. | $\alpha_{input}$ | $\alpha_{output}$ | LR decay | LR warm-up steps | Steps | Tokens |
|---|---|---|---|---|---|---|---|---|---|---|---|---|
| Fig. 1, 6, 8, 14 | 4096 | 2 | 64 | 128 | Variable | SP: 2.166E-2 / μP, SμPar: 0.087 | 9.1705 | 1.095 | 10x linear | 116 | 1169 | 306M |
| Fig. 7, 13 | 4096 | 2 | 64 | 8 | 1.62E-2 | Variable | 9.1705 | 1.095 | Constant | 0 | 100 | 1.6M |
| Fig. 3 | 2048 | 10 | 64 | 504 | SP: 2e-4 / μP, SμPar: 1.62E-2 | SP: 0.02 / μP, SμPar: 0.087 | 9.1705 | 1.095 | Decay to zero | 1175 | 11752 | 12.13B |
| Fig. 5 | 2048 | 2 | 32 | 4 | 1.68E-02 / μP, SμPar: 1.62E-2 | 0.101 | 11.22 | 1 | Constant | 0 | 10 | 82K |
| Fig. 9 | Variable | 2 | 64 | 128 | Variable | SP: 0.02 / μP, SμPar: 0.087 | 9.1705 | 1.095 | 10x linear | 116 | 1169 | 306M |
| Fig. 10 | Variable | 10 | 64 | 256 | SP, $d_{model} \leq 1088$: 6E-4 / SP, $d_{model} > 1088$: 2E-4 / μP, SμPar: 1.62E-2 | SP: 0.02 / μP, SμPar: 0.087 | 9.1705 | 1.095 | 10x linear | 190 | 1907 | 1B |
| Fig. 11 | Variable | 2 | 64 | 128 | Variable | 0.087 for SP. | N/A | N/A | 10x linear | 116 | 1169 | 306M |
| Fig. 12 | 1024 | 2 | 64 | 128 | Variable | SP: 2.166E-2 / μP, SμPar: 0.087 | 9.1705 | 1.095 | 10x linear | 116 | 1169 | 306M |

