# OpenReview forum: "Sparse maximal update parameterization: A holistic approach to sparse training dynamics"
_NeurIPS.cc/2024/Conference — NeurIPS 2024 poster_

### Official Review · Reviewer_Grts · 2024-07-03

**Soundness:** 3
**Presentation:** 2
**Contribution:** 3
**Rating:** 7
**Confidence:** 4

**Summary:**

This work (similarly to \muP method) proposes a specific parameterization for weights, gradients and updates such that the optimal training hyperparameters (i.e. learning rate) are transferrable across different sparsity levels. The approach is validated on the task of language modeling within wide range of sparsities.

**Strengths:**

* The proposed approach is theoretically sound and intuitive.

* SμPar appears to produce a stable optimal learning rate with a wide range of sparsities, whereas the values for standard parameterization and dense \muP have to be tuned according to the level of sparsity.

* SμPar allows one to achieve strong results for larger, outperforming the tuned SP and \muP configuration, given a small proxy tuned model. With significantly smaller tuning cost one can achieve better results.

**Weaknesses:**

Whereas all the components in SμPar seem to be important, the importance of each individual parameterization is not studied. It could be the case that SμPar for weights while keeping \muP parameterization for gradients and learning rate may suffice and produce the same performance. For the sake of completeness, I would suggest ablating individual terms.
Prior [1], [2] work shows that proper weight initialization (scaled according to sparsity) already improves training dynamics.

**Questions:**

Does the SμPar transfer to other domains in the same way? For example, one could try pretraining a vision transformer in a setup similar to [3].

Sparsity scaling laws paper [3] showed that given sufficiently long compute, sparse models turn out to be compute-optimal. Given that SμPar finds better hyperparameters, could one expect that higher sparsity levels may be favored compared to the recipe used in the aforementioned work?

It seems like the value of train loss decreases with the increase of sparsity (i.e., sparse models are better given the same number of parameters), whereas in Figure 10, the performance of SμPar loss seems to be independent of sparsity. I would expect an increase in the quality gap for longer training according to sparsity scaling laws.

---
[1] Evci, Utku, et al. "Gradient flow in sparse neural networks and how lottery tickets win." Proceedings of the AAAI conference on artificial intelligence. Vol. 36. No. 6. 2022.

[2] Liu, Zhuang, et al. "Rethinking the value of network pruning." arXiv preprint arXiv:1810.05270 (2018).

[3] Frantar, Elias, et al. "Scaling laws for sparsely-connected foundation models." arXiv preprint arXiv:2309.08520 (2023).

**Limitations:**

See Weaknesses and Questions.

Would be interesting to see whether the introduced parametrization is optimal for training methods with dynamic sparsity (RigL [1], Top-Kast [2], AC/DC [3]).

---
[1] Evci, Utku, et al. "Rigging the lottery: Making all tickets winners." International conference on machine learning. PMLR, 2020.

[2] Jayakumar, Siddhant, et al. "Top-kast: Top-k always sparse training." Advances in Neural Information Processing Systems 33 (2020): 20744-20754.

[3] Peste, Alexandra, et al. "Ac/dc: Alternating compressed/decompressed training of deep neural networks." Advances in neural information processing systems 34 (2021): 8557-8570.

---

> ### Author Rebuttal · Authors · 2024-08-07
>
> Thank you very much for your thoughtful review.  It is gratifying to know that you found the method theoretically sound and intuitive, and that the experimental results were a strength of the paper.
>
> ```
> Whereas all the components in S$\mu$Par seem to be important, the importance of each individual parameterization is not studied. It could be the case that S$\mu$Par for weights while keeping $\mu$P parameterization for gradients and learning rate may suffice and produce the same performance. For the sake of completeness, I would suggest ablating individual terms. Prior [1], [2] work shows that proper weight initialization (scaled according to sparsity) already improves training dynamics.
> ```
> This is an excellent suggestion! In Section 1 "Individual ablations of S$\mu$Par initialization and learning rate corrections" from our "Global Author Rebuttal" we perform the ablation you suggested and show that both the initialization and learning rate correction are required for S$\mu$Par to achieve transfer of optimal initialization standard deviation or optimal learning rate across sparsity levels. We will include these ablations in the camera version of the paper.
>
> ```
> Does the S$\mu$Par transfer to other domains in the same way? For example, one could try pretraining a vision transformer in a setup similar to "Scaling laws for sparsely-connected foundation models".
> ```
>
> This is a good question because much of the sparse neural network literature has focused on the vision domain. Since ViT shares most of its architecture with language transformers, we believe the extension of $\mu$P and S$\mu$Par to ViT should also follow Table 1 from our main submission, but we leave testing it as future work.
>
> However, the setup you are referencing also uses gradual magnitude pruning (GMP). In Section 2 "Dynamic sparsity hyperparameter transfer" from our "Global Author Rebuttal", we show that S$\mu$Par is not correct for GMP and provide an explanation as to why (continuing our explanation from Lines 274-278).
>
> ```
> "Scaling laws for sparsely-connected foundation models" showed that given sufficiently long compute, sparse models turn out to be compute-optimal. Given that S$\mu$Par finds better hyperparameters, could one expect that higher sparsity levels may be favored compared to the recipe used in the aforementioned work?
> ```
> This is a very interesting paper with potentially high impact. On Lines 262-266, we discuss how the setup used in "Scaling laws for sparsely-connected foundation models" adjusts the learning rate proportional to sparsity. This correction likely has a similar effect to the S$\mu$Par learning rate correction however without both an appropriate initialization and learning rate correction, optimal hyperparameter transfer is not guaranteed. Without HP transfer, it is likely that many of the models trained in this paper experience "de-tuned hyperparameters" as width and sparsity are varied. As we discuss in Section 2 "Dynamic sparsity hyperparameter transfer" from our "Global Author Rebuttal", if one could extend S$\mu$Par for arbitrary sparse algorithms, then that parameterization would likely improve the sparse models in the aforementioned work, especially at high sparsity levels. We leave the development of this method as future work, as stated on Line 278.
>
> ```
> It seems like the value of train loss decreases with the increase of sparsity (i.e., sparse models are better given the same number of parameters), whereas in Figure 10, the performance of S$\mu$Par loss seems to be independent of sparsity. I would expect an increase in the quality gap for longer training according to sparsity scaling laws.
> ```
> Iso-FLOP efficiency isn't the focus of our paper. However in Figure 10 we do include the iso-Parameter scaling tests similar to the scaling strategy in "Scaling laws for sparsely-connected foundation models" as well as several other papers. However "iso-Parameter" doesn't necessarily mean iso-FLOP because increasing $d_\text{model}$ also increases attention dot product FLOPs, which can't be sparsified. This increase in attention FLOPs is so significant that our 87.5\% sparse model from Figure 10 has just over double the training FLOPs of the dense baseline, with virtually unchanged loss. Unfortunately, "Scaling laws for sparsely-connected foundation models" did not model the effect of attention FLOPs in their scaling law functional forms, instead assuming that sparsity and parameter count alone are sufficient. We suspect that many of the sparse models in that paper use considerably more total training FLOPs than their dense baselines, making for an unfair comparison. It is unclear whether their results hold under more robust FLOP comparisons. For this submission though, we consider the work of beating dense FLOP efficiency with sparse training as future work. We will update our manuscript to reflect the subtleties of FLOP counting in sparse models.
>
> ```
> Would be interesting to see if S$\mu$Par is optimal for DST (eg. RigL, Top-KAST, AC/DC)
> ```
> This is a good suggestion for an experiment! Unfortunately S$\mu$Par is not optimal for DST methods. We demonstrate this and provide an explanation in Section 2 "Dynamic sparsity hyperparameter transfer" from our "Global Author Rebuttal". We leave the solution as future work.

---

> > ### Comment · Reviewer_Grts · 2024-08-07
> >
> > Thanks for your response and ablations. Most of my concerns were resolved, as well as the additional results are quite insightful. Therefore, I decide to increase the score.

---

### Official Review · Reviewer_hqe9 · 2024-07-07

**Soundness:** 3
**Presentation:** 3
**Contribution:** 3
**Rating:** 6
**Confidence:** 4

**Summary:**

A parameterization method is proposed to ensure that the sparse models with different weight sparsity share the same set of optimal hyperparameters. The sparse models with S$\mu$PaR achieve lower validation accuracy using the same hyperparameters the dense model uses.

**Strengths:**

- The paper is well-written and easy to follow. The experimental results support the claim well.
- The proposed method links the maximal update and sparse network training, which is an interesting found.

**Weaknesses:**

- The proposed method is discussed and evaluated only with the random unstructured sparsity pattern, which is hardly used in practice to speed up the DNN training and inference.
- The theoretical contribution is somewhat weak. The proposed parameterization method is simple extension of maximal update parameterization to the sparse version, which I don't see significant technical improvement.
- The experimental results should include the accuracy comparison between S$\mu$PaR and other sparse weight initialization methods (e.g., [1]). Although the proposed method was compared with SP and $\mu$P, the superiority of S$\mu$PaR is somewhat predictable since the other two are not designed for the sparse weights. I am curious how much improvement S$\mu$PaR can make from the previous sparse weight initialization schemes.
- The scaling factors for the weight update need to be derived for each optimizer, which might limit the usability of S$\mu$PaR in practice.

[1] Lee, Namhoon, et al. "A signal propagation perspective for pruning neural networks at initialization." arXiv preprint arXiv:1906.06307 (2019).

**Questions:**

Major questions
- Is the proposed parameterization method generalizable to any structured sparsity patterns (e.g., channel pruning) for the density-invariant hyperparameters? As mentioned in the paper, the unstructured sparsity is extremely hard to speed up. Discussing S$\mu$PaR on more structured sparsity patterns, such as channel pruning, will make the paper stronger.
- Why does S$\mu$PaR stabilize the sparse network training even after multiple training steps? How is S$\mu$PaR technically different from the previous sparse weight initialization methods? Also, what is the accuracy difference between S$\mu$PaR and other sparse weight initialization methods?

Minor questions
- Line 113 about the weight update $\mathbf{X}^{l+1} + \Delta \mathbf{X}^{l+1}$ is confusing because the weight update is denoted by $\Delta \mathbf{W}^l$ in Line 105.
- Is there any connection between the proposed method and the iterative magnitude pruning (IMP)used in Lottery Ticket Hypothesis [2]? Can S$\mu$PaR improve the final performance of IMP?

[2] Frankle, Jonathan, and Michael Carbin. "The lottery ticket hypothesis: Finding sparse, trainable neural networks." arXiv preprint arXiv:1803.03635 (2018).

**Limitations:**

The authors have addressed the limitations and broader impacts in the paper.

---

> ### Author Rebuttal · Authors · 2024-08-07
>
> Thank you for your detailed and constructive feedback, and your kind words regarding the paper’s writing, experimental results, and core idea.
>
> ```
> SuPar is discussed and evaluated only with the random unstructured sparsity pattern, which is hardly used in practice to speed up the DNN training and inference.
> ```
> It is true that the current adoption rate of unstructured sparsity in training and inference is low, as we acknowledge on Lines 35-36 and 73-75 of our submission. While we currently mostly motivate unstructured sparse training on Lines 27-28 and 279-286, we will note in the paper introduction that unstructured sparsity remains a very important technique and intense focus of research for ourselves and other labs, as accelerators such as CPUs, Cerebras, GraphCore, and SambaNova can take advantage of unstructured sparsity during training and inference. Furthermore, GPU and TPU systems support 2:4 block structured sparsity which is quite similar to 50\% unstructured sparsity.
>
> Research has shown that the less structure imposed on sparsity masks, the better the performance, making this setting relevant to study in training and inference. For example, SparseGPT [Frantar and Alistarh(2023)], the SOTA LLM pruning technique creates unstructured sparse masks that require hardware acceleration to realize an inference benefit. They report a CPU inference speedup, demonstrating the potential for real-world impact of unstructured sparsity.
>
> Additionally, in Section 2 "Dynamic sparsity hyperparameter transfer" from our "Global Author Rebuttal", we test RigL and GMP dynamic sparse training and show that none of SP, $\mu$P, or S$\mu$Par achieve transfer of optimal learning rate across sparsity level. We hope this helps address your concerns around applying S$\mu$Par outside the context of random unstructured sparsity.
>
> ```
> The theoretical contribution is somewhat weak. The proposed parameterization method is simple extension of maximal update parameterization to the sparse version, which I don't see significant technical improvement.
> ```
> It is true that S$\mu$Par is a simple extension of $\mu$P. However, we note that we are the first to acknowledge and propose a solution to the systematic misuse of dense hyperparameters in sparse training. On Lines 90-91 and Footnote 1 (page 3) from our submission, we demonstrate how widespread the issue is by providing numerous examples of sparse training works which use dense hyperparameters for sparse training. So, regardless of the perceived significance of technical improvement, the work is important and the potential for impact is fairly high.
>
> ```
> Why does S$\mu$Par stabilize the sparse network training even after multiple training steps? How is S$\mu$Par technically different from the previous sparse weight initialization methods?
> ```
> We take your point, and indeed we discuss this on Lines 267-273. $\mu$P and S$\mu$Par differ from pure initialization-based methods due to the presence of a learning rate correction. The combination of an initialization and learning rate correction allows S$\mu$Par to stabilize sparse network training even after multiple training steps, as seen in Figure 5 of our main submission. In Section 1 "Individual ablations of S$\mu$Par initialization and learning rate corrections" from our "Global Author Rebuttal", we also show that the S$\mu$Par initialization adjustment alone is not sufficient to achieve transfer of optimal HPs across sparsity levels.
>
> Additionally, please see Section 3 "Downstream Task Evaluations" from our "Global Author Rebuttal" where we provide accuracy comparisons between SP, $\mu$P, and S$\mu$Par.
>
> We also think the comparison across PaI methods is somewhat orthogonal to our work because we are not studying the optimal sparse training recipe (eg. PaI, DST, prune-retrain), but instead how to best control sparse training dynamics.
>
> ```
> Is the proposed parameterization method generalizable to any structured sparsity patterns (e.g., channel pruning) for the density-invariant hyperparameters? As mentioned in the paper, the unstructured sparsity is extremely hard to speed up. Discussing S$\mu$Par on more structured sparsity patterns, such as channel pruning, will make the paper stronger.
> ```
>
> Great question! On Lines 160-161 we remark that S$\mu$Par should easily extend to random 2:4 structured sparsity as it closely resembles random unstructured sparsity. S$\mu$Par would also work for random structured pruning of entire neurons at initialization because this case simply reduces to training with a narrower dense model. However, S$\mu$Par does not transfer to dynamic sparse training because weights become non-Gaussian. Please see Section 2 "Dynamic sparsity hyperparameter transfer" from our "Global Author Rebuttal" for a discussion on the limitations of S$\mu$Par in dynamic sparse training. We will update the main body to more thoroughly discuss these possible extensions. Thank you for the feedback.
>
> ```
> Can S$\mu$Par improve the final performance of IMP?
> ```
> This is an interesting question. S$\mu$Par improves random static unstructured sparse training and IMP rewinds weights back to their initial values while maintaining the same mask. If the IMP mask at initialization still allows the non-zero weights to have a Gaussian distribution, then S$\mu$Par would apply to this case. Therefore, S$\mu$Par *could* prevent ``hyperparameter detuning'' in later IMP iterations, and *potentially* improve IMP losses. Given the popularity of IMP/LTH, we think this discussion would make an excellent addition to Section 5 of our paper. Thank you for bringing this up!
>
> ```
> DeltaX vs DeltaW notation
> ```
> $\Delta W$ refers to the weight update whereas $\Delta \mathbf{X}$ refers to "the effect of the weight update”. We will update Line 113 accordingly.
>
> [Frantar and Alistarh(2023)] Elias Frantar and Dan Alistarh. 2023. SparseGPT: Massive language models can be accurately pruned in one-shot. In International Conference on Machine Learning.

---

> > ### Comment · Reviewer_hqe9 · 2024-08-08
> >
> > Thanks for your thorough response to my review. My primary concern was the practicality of S$\mu$PaR and unstructured sparse weights in general, which was effectively resolved by the authors' response. Also, the additional experimental results are impressive. I am inclined to raise my score.

---

### Official Review · Reviewer_6wqj · 2024-07-12

**Soundness:** 2
**Presentation:** 3
**Contribution:** 3
**Rating:** 5
**Confidence:** 4

**Summary:**

The paper introduces Sparse Maximal Update Parameterization (SµPar), a novel approach designed to address challenges in training sparse neural networks. Sparse networks, despite their computational advantages, often struggle with signal propagation and optimal hyperparameter (HP) tuning across different sparsity levels. SµPar aims to stabilize training dynamics by ensuring that activations, gradients, and weight updates are scale-invariant with respect to sparsity. This method allows for consistent optimal HPs across varying model widths and sparsity levels, thus reducing the tuning costs associated with sparse models. The empirical results demonstrate that SµPar outperforms standard parameterization (SP) and maximal update parameterization (µP) in maintaining stable optimal HPs and achieving better loss metrics across different sparsity levels and model scales.

**Strengths:**

1. The idea of this paper is interesting. The authors select a trending topic with very clear and strong motivation. The preliminary experimental results is very clear and helpful to the readers.

2. The discussion about forward/backward pass and weight updates are very clear. Figures are also clear and good to read.

3. The writing is good and the paper is well structured.

**Weaknesses:**

1. This paper has many supporting experiments reporting the training loss, validation loss, and transfer loss for SµPar. However, loss values are not always reliable for evaluation. The accuracy data is a more convincing data to demonstrate the performance. I wonder what is the accuracy for the experiments. It would make reader easy to compare against other methods.

2. What is the method that is used to prune the network? There are lots of method such as SNIP and GraSP for static sparse training. However, this part is unclear according to the paper.

3. The author claims SµPar is a holistic solution. However, from what the paper has demonstrate (discussion, experiments), the reviewer cannot be convinced. According to weaknesses 2, we don’t know what is the pruning method used in sparse training, and according to the paper, the author also not performs applying SµPar on different sparse training methods. Therefore, we don’t know if SµPar works for different sparse training algorithms. Furthermore, I assume this paper only discuss static sparse training (such as SNIP), without discussing the performance on dynamic sparse training (DST) which is a more reliable and promising sparse training domain (e.g., better accuracy, flexible). Therefore it is not suitable to claim SµPar is a holistic solution. Even the authors don’t make such claim, without sufficient experiments the reviewer mentioned in this part, this paper is still lacks fundamental evidence to demonstrate effectiveness.

**Questions:**

Please refer to weaknesses.

**Limitations:**

Please refer to weaknesses.

---

> ### Author Rebuttal · Authors · 2024-08-07
>
> Thank you very much for the helpful comments, and for your kind words regarding the paper's key idea, motivation, and clear and well-structured presentation.
>
> ```
> This paper has many supporting experiments reporting the training loss, validation loss, and transfer loss for SµPar. However, loss values are not always reliable for evaluation. The accuracy data is a more convincing data to demonstrate the performance. I wonder what is the accuracy for the experiments. It would make reader easy to compare against other methods.
> ```
>
> Thank you for suggesting this. Please see Section 3 "Downstream Task Evaluations" from our "Global Author Rebuttal". We hope this addresses your concerns.
>
> ```
> What is the method that is used to prune the network? There are lots of method such as SNIP and GraSP for static sparse training. However, this part is unclear according to the paper.
> ```
> Thank you for bringing this lack of clarity to our attention! In this work we solely used random static unstructured pruning at initialization, as stated on Lines 107, 512, 528, 549, and 564. To improve clarity we will update our abstract, intro, and conclusion to further emphasize our choice of pruning method. As we noted on Lines 27-28, recent work has shown random sparsity to be a surprisingly effective strategy for sparse training, so we adopted this in our work.
>
> ```
> The author claims SµPar is a holistic solution. However, from what the paper has demonstrate (discussion, experiments), the reviewer cannot be convinced. According to weaknesses 2, we don’t know what is the pruning method used in sparse training, and according to the paper, the author also not performs applying SµPar on different sparse training methods. Therefore, we don’t know if SµPar works for different sparse training algorithms. Furthermore, I assume this paper only discuss static sparse training (such as SNIP), without discussing the performance on dynamic sparse training (DST) which is a more reliable and promising sparse training domain (e.g., better accuracy, flexible). Therefore it is not suitable to claim SµPar is a holistic solution. Even the authors don’t make such claim, without sufficient experiments the reviewer mentioned in this part, this paper is still lacks fundamental evidence to demonstrate effectiveness.
> ```
>
> Thank you for raising these points. Please see Section 2 "Dynamic sparsity hyperparameter transfer" from our "Global Author Rebuttal". This section shows that none of SP, $\mu$P, or S$\mu$Par achieve transfer of optimal learning rate across sparsity level for RigL and GMP. We hope this addresses your concerns around applying S$\mu$Par to dynamic sparse training methods.
>
> Also, we agree that ``holistic'' is a vague term, as it might imply S$\mu$Par solves challenges with other forms of sparse training, such as dynamic sparse training.  We should have been more clear that S$\mu$Par holistically controls the scale of the forward, backward, and weight update operations with respect to model width and level of unstructured static random sparsity –- some of these operations have been handled \emph{in isolation} in prior work, but not all together.
>
> We will also update our introduction to make it clear that S$\mu$Par does not extend to pruning or dynamic sparse training because these methods can cause the unmasked/non-zero weight distribution to become non-Gaussian (as noted currently only in L274-L278).

---

> > ### Comment · Reviewer_6wqj · 2024-08-13
> > **Thank you for your rebuttal**
> >
> > Dear authors,
> >
> > Thank you for your rebuttal and additional experiments. It seems that the performance on DST is not that good. But most of my concerns are addressed. I increase my score to 5

---

### Official Review · Reviewer_F4Ay · 2024-07-19

**Soundness:** 2
**Presentation:** 1
**Contribution:** 2
**Rating:** 3
**Confidence:** 3

**Summary:**

This paper studies the effect of various hyperparameters on static sparse training while having a holistic approach. It highlights that there is a correlation between their settings and neural network performance (practically loss function values). The experiments are performed on smaller and larger models, including a large-scale language model.

**Strengths:**

This is an interesting study with a relatively low level of originality (in my opinion). It practically puts together and discuss a bit more various findings from various papers. Also, the paper structure and clarity could have been improved. Currently, it is not easy to understand it. If the paper would be more mature, may have a relatively fair impact in the community.

**Weaknesses:**

I believe that one main weakness is that the networks performance is reported just with loss values. For a more comprehensive view, and deeper understanding other performance metrics (task dependent) shall be also presented for all experiments.

With respect to the problem motivation, I believe that the paper exaggerates a bit the difficulty of training static sparse neural networks. Perhaps, for static sparsity, the effect of hyperparameters on network performance is more pronounced, but the alternative dynamic sparsity is quite robust to hyperparameter choice (e.g., sparsity level) and can obtain quite easy performance on par (or even better) with dense neural networks while having much fewer theoretical computational requirements as reported in a number of related works. For a fair overview, the paper shall discuss also comparatively the performance of static sparsity against dynamic sparsity and pruning, or alternatively, to motivate better its settings.

The paper readability can be improved with a better structure, more clarity in arguments and qualitative discussions, and more clear statements about the paper novel contributions.

(minor) The source code shall be provided to ensure easy reproducibility.

**Questions:**

I don't have questions which may change my evaluation. Still, I consider this study promising, and I hope that my comments will help the authors to improve the next version of the paper.

---

> ### Author Rebuttal · Authors · 2024-08-07
>
> Thank you for your very useful suggestions, your positive comments regarding the promising/interesting findings, and your note on the potential for impact in the community.
>
> ```
> This is an interesting study with a relatively low level of originality (in my opinion). It practically puts together and discuss a bit more various findings from various papers. Also, the paper structure and clarity could have been improved. Currently, it is not easy to understand it. If the paper would be more mature, may have a relatively fair impact in the community.
> ```
> We take your point regarding the level of originality, however we note that we are the first to acknowledge and propose a solution to the systematic misuse of dense hyperparameters in sparse training. On Lines 90-91 and Footnote 1 (page 3) from our submission, we demonstrate how widespread the issue is by providing numerous examples of sparse training works which use dense hyperparameters for sparse training. So, regardless of the perceived level of innovation, the work is important and the potential for impact is fairly high, as you noted.
>
> ```
> I believe that one main weakness is that the networks performance is reported just with loss values. For a more comprehensive view, and deeper understanding other performance metrics (task dependent) shall be also presented for all experiments.
> ```
>
> Thank you for suggesting this. Please see Section 3 "Downstream Task Evaluations" from our "Global Author Rebuttal". We hope this addresses your concerns.
>
> ```
> With respect to the problem motivation, I believe that the paper exaggerates a bit the difficulty of training static sparse neural networks. Perhaps, for static sparsity, the effect of hyperparameters on network performance is more pronounced, but the alternative dynamic sparsity is quite robust to hyperparameter choice (e.g., sparsity level) and can obtain quite easy performance on par (or even better) with dense neural networks while having much fewer theoretical computational requirements as reported in a number of related works. For a fair overview, the paper shall discuss also comparatively the performance of static sparsity against dynamic sparsity and pruning, or alternatively, to motivate better its settings.
> ```
> This is very perceptive –- and a point that we did not fully appreciate previously.
> But first of all, we should have done a better job of motivating static random sparsity – the topic of the paper – to begin with.  We will note in the paper that static sparsity remains a very important technique and intense focus of research for ourselves and other labs, not least because it enjoys native hardware support (e.g., 2:4 sparsity in NVIDIA GPUs, full unstructured sparsity in Cerebras chips).
>
> That being said, investigating hyperparameter shift in DST methods is insightful. Please see Section 2 "Dynamic sparsity hyperparameter transfer" from our "Global Author Rebuttal". In this section we show that the optimal learning rate significantly changes across sparsity levels for both RigL and GMP, contradicting your claim that dynamic sparsity is quite robust to hyperparameter choice and further highlighting the importance of developing systematic solutions like S$\mu$Par. As we discussed on Lines 274-278 of our submission, S$\mu$Par is not correct for dynamic sparse methods, leaving an impactful direction for future work.
>
> We really appreciate your suggestion here as these interesting findings definitely help expand the paper’s contribution!
>
> ```
> (minor) The source code shall be provided to ensure easy reproducibility.
> ```
> Unfortunately we do not have an open-source repository for S$\mu$Par. However, we do provide Table 1 in our submission, which outlines the simple implementation changes required to implement $\mu$P and S$\mu$Par. Based on this it should be straightforward to extend existing implementations of $\mu$P to implement S$\mu$Par (e.g. https://github.com/microsoft/mup).

---

> > ### Comment · Reviewer_F4Ay · 2024-08-12
> > **Thank you for the rebuttal**
> >
> > Dear authors,
> >
> > Thank you for considering my comments and for preparing an extensive rebuttal. I still believe that all these (including your answers to the other reviewers) has to be integrated in a new version of the paper and that version will still need a new peer-review process. Therefore, I have to keep my original rating. Nevertheless, I take your point about the systematic misuse of dense hyperparameters in sparse training as novel indeed, and for this reason, I will not put myself against acceptance if the other reviewers and the area chair will unanimously support the acceptance of this paper.
> >
> > Best wishes,
> > Reviewer F4Ay

---

> > > ### Author Response · Authors · 2024-08-13
> > > **Thank you for the response**
> > >
> > > Thanks a lot for engaging in this discussion and for being open to acceptance based on the novel contributions.  Given the other reviewers are now unanimous in recommending acceptance, it would probably clarify things for the AC if your score also indicated you were no longer standing against acceptance.  However what eventually matters is the review text, not the final score, so thanks again either way!

---

### Author Rebuttal · Authors · 2024-08-07

We thank all reviewers for taking the time to read our submission and provide helpful feedback. Please find attached our 1 page PDF containing additional results. We believe these additions help address many reviewer concerns and strengthen our submission. Here we provide a discussion of the results in our 1 page Rebuttal PDF.

# Section 1: Individual ablations of S$\mu$Par initialization and learning rate corrections
Thank you to reviewer Grts for suggesting we individually ablate the effect of the S$\mu$Par initialization and the S$\mu$Par learning rate. In Figure 1 of our Rebuttal PDF (blue and orange boxes), we show that using only the S$\mu$Par initialization in conjunction with $\mu$P ("$\mu$P + S$\mu$Par init only") does not allow for transfer of optimal initialization standard deviation or optimal learning rate across sparsity levels. This result also helps address some of the feedback from reviewer hqe9. We also show that using only the S$\mu$Par learning rate in conjunction with $\mu$P does not achieve transfer either ("$\mu$P + S$\mu$Par LR only"). Therefore, both the S$\mu$Par initialization and learning rate corrections are required to achieve optimal hyperparameter transfer across sparsity levels.

# Section 2: Dynamic sparsity hyperparameter transfer
Every reviewer mentioned dynamic sparsity in some capacity, which motivated us to study it more closely in the context of S$\mu$Par. In Figure 1 of our Rebuttal PDF (green box), we test the transfer of optimal learning rate across sparsity levels for two popular dynamic sparse training methods: Rigging the Lottery (RigL) [Evci et al.(2020)] and Gradual Magnitude Pruning (GMP) [Zhu and Gupta(2017)]. We show that none of SP, $\mu$P, or S$\mu$Par achieve transfer of optimal learning rate across sparsity levels. For SP and $\mu$P we see that higher sparsity levels have higher optimal learning rates. This is because sparsity reduces activation and gradient scales such that a larger learning rate is needed to counteract this. S$\mu$Par sees the opposite trend where higher sparsity levels have lower optimal learning rates, indicating that S$\mu$Par is ``overcorrecting''.

On Lines 274-278 of our submission we mention that dynamic sparse methods can make updates to the weight mask such that the distribution of unmasked/non-zero weights changes to something non-Gaussian, which prevents S$\mu$Par from being mathematically correct. Compared to random pruning, a mask obtained from magnitude pruning will better preserve the size of activations and gradients seen in the dense network. Since S$\mu$Par assumes weights are drawn from a Gaussian distribution, S$\mu$Par ends up ``overcorrecting'' the initialization and learning rate. In future work it would be impactful to develop a parameterization which generalizes S$\mu$Par to work for an arbitrary sparse training algorithm.

# Section 3: Downstream Task Evaluations
Thank you to reviewer F4Ay and 6wqj for pointing out the limitations of limitations of relying on loss alone for evaluating LLMs. We recognize this shortcoming since ultimately LLMs are being trained for use in downstream tasks. Following their suggestion, in Table 1 of the Rebuttal PDF, we evaluated the models from Figure 1 of our submission to provide a head-to-head comparison between SP, $\mu$P, and S$\mu$Par. Results across pretraining loss and average downstream task accuracy consistently show that S$\mu$Par models achieve superior performance compared to SP and $\mu$P. We measured accuracy on five downstream tasks: ARC-easy, lambada, RACE, PIQA, and BoolQ, which collectively test for common sense reasoning, world knowledge, and reading comprehension. We also specifically chose tasks that are easy enough for even extremely sparse models to significantly outperform random chance.

[Evci et al.(2020)] Utku Evci, Trevor Gale, Jacob Menick, Pablo Samuel Castro, and Erich Elsen. 2020. Rigging the lottery: Making all tickets winners. In International conference on machine learning. PMLR, 2943–2952.

[Zhu and Gupta(2017)] Michael Zhu and Suyog Gupta. 2017. To prune, or not to prune: exploring the efficacy of pruning for model compression. arXiv preprint arXiv:1710.01878 (2017).

---

### Decision · Program_Chairs · 2024-09-25

**Decision:**

Accept (poster)

**Comment:**

This paper proposes a specific parameterization for weights, gradients and updates such that the optimal training hyperparameters (i.e. learning rate) remain the same across different sparsity levels or widths. The approach is validated on the task of language modeling across various levels of sparsities. Some main concerns about the novel contributions, evaluation metrics (loss), and dynamic sparse training, etc. During the rebuttal period, the authors clarify with new results. Most of reviewers are satisfied with the authors rebuttal and propose to accept the paper. Although one reviewer still thinks the paper needs more polishing work with another round of review, this reviewer also does not object to accept if all the others agree to accept the paper. Therefore, we propose to accept the paper, suggesting the authors to include all the clarifications and new results in the final version.